# Attainability and Optimality: The Equalized Odds Fairness Revisited

**Zeyu Tang**                                        ZEYUTANG@CMU.EDU
*Carnegie Mellon University*

**Kun Zhang**                                        KUNZ1@CMU.EDU
*Carnegie Mellon University, and*
*Mohamed bin Zayed University of Artificial Intelligence*

**Editors:** Bernhard Schölkopf, Caroline Uhler and Kun Zhang

## Abstract

Fairness of machine learning algorithms has been of increasing interest. In order to suppress or eliminate discrimination in prediction, various notions as well as approaches have been proposed to impose fairness. Given a notion of fairness, an essential problem is then whether or not it can always be attained, even if with an unlimited amount of data. This issue is, however, not well addressed yet. In this paper, focusing on the Equalized Odds notion of fairness, we consider the attainability of this criterion and, furthermore, if it is attainable, the optimality of the prediction performance under various settings. In particular, for prediction performed by a deterministic function of input features, we give conditions under which Equalized Odds can hold true; if the stochastic prediction is acceptable, we show that under mild assumptions, fair predictors can always be derived. For classification, we further prove that compared to enforcing fairness by post-processing, one can always benefit from exploiting all available features during training and get potentially better prediction performance while remaining fair. Moreover, while stochastic prediction can attain Equalized Odds with theoretical guarantees, we also discuss its limitation and potential negative social impacts.

**Keywords:** Algorithmic Fairness, Equalized Odds, Conditional Independence, Attainability, Optimality

## 1. Introduction

As machine learning models become widespread in automated decision making systems, apart from the efficiency and accuracy of the prediction, their potential social consequence also gains increasing attention. The call for accountability and fairness in machine learning has motivated various (statistical) notions of fairness. *Demographic Parity* (Calders et al., 2009) requires the independence between prediction (e.g., of a classifier) and the protected feature (sensitive attributes of an individual, e.g., gender, race). *Equalized Odds* (Hardt et al., 2016), also known as *Error-rate Balance* (Chouldechova, 2017), requires the output of a model be conditionally independent of protected feature(s) given the ground truth of the target. *Predictive Rate Parity* (Zafar et al., 2017a), on the other hand, requires the actually proportion of positives (negatives) in the original data for positive (negative) predictions should match across groups (well-calibrated).

On the theoretical side, results have been reported regarding relationships among fairness notions. It has been shown that if base rates of positives differ among groups, then Equalized Odds and Predictive Rate Parity cannot be achieved simultaneously for non-perfect predictors (Kleinberg et al., 2017; Chouldechova, 2017). Any two out of three among Demographic Parity, Equalized Odds, and Predictive Rate Parity are incompatible with each other (Barocas et al., 2017).

In practice, one can broadly categorize computational procedures to derive a fair predictor into three types: pre-processing approaches (Calders et al., 2009; Dwork et al., 2012; Zemel et al., 2013; Zhang et al., 2018; Madras et al., 2018; Creager et al., 2019; Zhao et al., 2020), in-processing approaches (Kamishima et al., 2011; Pérez-Suay et al., 2017; Zafar et al., 2017a,b; Donini et al., 2018; Song et al., 2019; Mary et al., 2019; Baharlouei et al., 2020; Romano et al., 2020), and post-processing approaches (Hardt et al., 2016; Fish et al., 2016; Dwork et al., 2018). In accord with the fairness notion of interest, a pre-processing approach first maps the training data to a transformed space to remove discriminatory information between protected feature and target, and then pass on the data to make prediction. In direct contrast, a post-processing approach treats the off-the-shelf predictors (usually limited to classification tasks) as uninterpretable black-box(es), and imposes fairness by outputting a function of the original prediction. For in-processing approaches, various kinds of regularization terms are proposed so that one can optimize the utility function while suppressing the discrimination at the same time. Approaches based on estimating/bounding the causal effect of the protected feature on the final target have also been proposed (Kusner et al., 2017; Russell et al., 2017; Zhang et al., 2017; Nabi and Shpitser, 2018; Zhang and Bareinboim, 2018; Chiappa, 2019; Wu et al., 2019).

The attainability of Equalized Odds, namely, the existence of the predictor that can score zero violation of fairness in the large sample limit, is an asymptotic property of the fairness criterion that has important practical implications. It characterizes a completely different kind of violation of fairness compared to the empirical error bound of discrimination in finite-sample cases. If we deploy a "fair" prediction system in practice whose output is actually biased (because fairness is in fact not attainable with the chosen predictor), the discrimination would become a snake in the grass, which is often easily neglected and hard to eliminate. Although various approaches have been proposed to impose Equalized Odds, whether or not it is always attainable is not well addressed. Hardt et al. (2016) already noticed that one "might need to introduce additional randomness [to achieve Equalized Odds in binary classification]". Compared to the claim that "[deterministic classifier] might not be sufficient [to achieve fairness]" in the FICO score case study by Hardt et al. (2016), we present an affirmative answer with respect the necessary and sufficient conditions under which Equalized Odds can be attained (for both regression and classification tasks). Actually, as we illustrate in this paper, Equalized Odds is not always attainable if we use deterministic prediction functions. Our contributions are mainly threefold:

- For regression and classification tasks with deterministic prediction functions, we show that Equalized Odds is not always attainable unless certain (rather restrictive) conditions on the joint distribution of the features and the target variable are met.
- For regression and (binary) classification, we show that with a stochastic predictor, under mild assumptions one can always derive a non-trivial Equalized Odds predictor.
- Considering the optimality of performance under fairness constraint(s), when exploiting all available features, we show that the predictor derived via an in-processing approach would generally perform better, and at least no worse than the one derived via a post-processing approach (unconstrained optimization followed by a post-processing step).

## 2. Preliminaries

In this section, we first distinguish between objects of interest with respect to which fairness is discussed, and then present the definition of Equalized Odds.

## 2.1. Different Fairness Semantics

Before presenting the fairness formulation, it is helpful to see the distinction between different semantics of fairness when discussing fair predictors. When evaluating the performance of the proposed fair predictor, it is a common practice to compare the loss (with respect to the utility function of choice, e.g., accuracy for binary classification) computed on target variable and the predicted value. There is an implicit assumption behind this practice: the generating process of the data, which is just describing a real-world procedure, is *not* biased in any sense (Danks and London, 2017). Only when we treat the target variable (recorded in the dataset) as unbiased can we justify the practice of loss evaluation and the conditioning on target variable when imposing fairness (as we shall see in the definition of Equalized Odds in Equation 1).

One may consider a music school admission example. The music school committee would decide if they admit a student to the violin performance program based on the applicant's personal information, educational background, instrumental performance, and so on. When evaluating whether or not the admission is "fair", the semantics of fairness come in at least two folds. First, based on the information at hand, did the committee evaluate the qualification of applicants without bias (How committee evaluate the applicants)? Second, is committee's procedure of evaluating applicants' qualification reasonable (How other people view the evaluation procedure used by the committee)?

In this paper, we assume the data recorded is unbiased and focus on the prediction (made with respect to current reality), such that the prediction itself does not include any biased utilization of information. The fairness with respect to the data generating procedure as well as the potential future influence of the prediction are beyond the scope of this paper.

## 2.2. Equalized Odds Fairness

Hardt et al. (2016) proposed *Equalized Odds* which requires the conditional independence between prediction and protected feature given ground truth of the target. Let us denote the protected feature by $A$, with domain of value $\mathcal{A}$, additional (observable) features by $X$, with domain of value $\mathcal{X}$, target variable by $Y$, with domain $\mathcal{Y}$, (not necessarily fair) predictors by $\widehat{Y}$, and fair predictors by $\widetilde{Y}$. Equalized-Odds fairness requires

$$\widetilde{Y} \perp\!\!\!\perp A \mid Y. \tag{1}$$

For classification tasks, one can conveniently use the probability distribution form, i.e., $\forall a \in \mathcal{A}, y, \tilde{y} \in \mathcal{Y}$:

$$P(\widetilde{Y} = \tilde{y} \mid A = a, Y = y) = P(\widetilde{Y} = \tilde{y} \mid Y = y), \tag{2}$$

$$\text{or more concisely, } P_{\widetilde{Y}|AY}(\tilde{y} \mid a, y) = P_{\widetilde{Y}|Y}(\tilde{y} \mid y). \tag{3}$$

For better readability, we also use the formulation in Equation 3 in cases without ambiguity. In the context of binary classification ($\mathcal{Y} = \{0, 1\}$), Equalized Odds requires that the True Positive Rate (TPR) and False Positive Rate (FPR) of each certain group match TPR and FPR of the population. Throughout the paper, without loss of generality we assume that there is only one protected feature for the purpose of simplifying notation. However, considering the fact that the protected feature can be discrete (e.g., race, sex) or continuous (e.g., the ratio of ethnic group in the population for certain district of a city), we do not assume discreteness of the protected feature. Due to the space limit, we will focus on the illustration and implication of our results and defer all the proofs to Section A of the appendix.

## 3. Fairness in Regression

Various regularization terms have also been proposed to suppress discrimination when predicting a continuous target (Berk et al., 2017; Zhang et al., 2018; Mary et al., 2019; Romano et al., 2020). However, whether or not one can always achieve 0-discrimination for regression, even if with an unlimited amount of data, is not clear. In this section we first consider a simple setup with linearly correlated continuous data as an example to show that Equalized Odds is not always attainable. Then we consider more general cases where regression is performed by a deterministic predictor and derive the condition under which Equalized Odds can hold true. Finally, when stochastic prediction is utilized, we show that under mild assumptions one can always find a non-trivial fair predictor, i.e., Equalized Odds is guaranteed to be (non-trivially) attainable.

### 3.1. Unattainability of Equalized Odds in Linear Non-Gaussian Regression

Let us start with the specific linear, non-Gaussian situation where the data is generated as follows ($H$ is not measured in the dataset):

$$X = qA + E_X, \;\; H = bA + E_H, \;\; Y = cX + dH + E_Y, \tag{4}$$

where $(A, E_X, E_H, E_Y)$ are mutually independent and $q, b, c, d$ are constants.

Let us denote $E := E_Y + dE_H$ and let $\widehat{Y}$ be a linear combination of $A$ and $X$, i.e., $\widehat{Y} = \alpha A + \beta X = (\alpha + q\beta)A + \beta E_X$, with linear coefficients $\alpha$ and $\beta$, where $\beta \neq 0$. In Theorem 1, we present the general result in linear non-Gaussian cases, where one cannot achieve the conditional independence between $\widehat{Y}$ and $A$ given $Y$.

**Theorem 1** *(Unattainability of Equalized Odds in the Linear Non-Gaussian Case)*
*Assume that $X$ has a causal influence on $Y$, i.e., $c \neq 0$ in Equation 4, and that $A$ and $Y$ are not independent, i.e., $qc + bd \neq 0$. Assume $p_{E_X}$ and $p_E$ are positive on $\mathbb{R}$. Let $f_1 := \log p_A$, $f_2 := \log p_{E_X}$, and $f_3 := \log p_E$. Further assume that $f_2$ and $f_3$ are third-order differentiable. Then if at most one of $E_X$ and $E$ is Gaussian, $\widehat{Y}$ is always conditionally dependent on $A$ given $Y$.*

From Theorem 1, we can see that if at most one of $E_X$ and $E$ is Gaussian, then any linear combination of $A$ and $X$ with non-zero coefficients will not be conditionally independent from $A$ given $Y$, meaning that it is impossible to (non-trivially) achieve Equalized Odds with a linear model in the linear non-Gaussian case.

### 3.2. Regression with Deterministic Prediction

We have seen that in the linear non-Gaussian case, any non-zero linear combination of the feature (which is a deterministic prediction function of the input) will not satisfy Equalized Odds. One may naturally wonder, instead of only considering linear models, whether or not Equalized Odds can be achieved in general regression cases. It is therefore desirable to derive the condition under which Equalized Odds can hold true for regression with deterministic prediction functions.

**Theorem 2** *(Condition to Achieve Equalized Odds for Regression with Deterministic Prediction)*
*Assume that the protected feature $A$ and the continuous target variable $Y$ are dependent and that their joint probability density $p(A, Y)$ is positive for every combination of possible values of $A$ and $Y$. Further assume that $Y$ is not fully determined by $A$, and that there are additional features $X$*

*that are not independent of $Y$. Let the prediction $\widehat{Y}$ be characterized by a deterministic function $f : \mathcal{A} \times \mathcal{X} \to \mathcal{Y}$. Equalized Odds holds true if and only if the following condition is satisfied ($\delta(\cdot)$ is the delta function):*

$$\forall y, \hat{y} \in \mathcal{Y}, \ \forall a, a' \in \mathcal{A}, a \neq a' : \ Q(a, y, \hat{y}) = Q(a', y, \hat{y}),$$

$$\text{where} \ \ Q(a, y, \hat{y}) \stackrel{\triangle}{=} \int_{\mathcal{X}} \delta\big(\hat{y} - f(a, x)\big) p_{X|AY}(x|a, y) dx.$$

In special cases when $X \perp\!\!\!\perp A \mid Y$ and $f$ is a function of only $X$ (e.g., a linear function with $0$ coefficient for $A$), this condition is always satisfied. In general cases, however, this condition specifies a rather strong constraint on the relation between the conditional probability density $p_{X|AY}(x \mid a, y)$ and the function $f$. Because of the integration over the product of $\delta(\cdot)$ function and $p_{X|AY}(x \mid a, y)$, the aforementioned condition requires $f$ to "pick out" certain $x$'s (for any given $\hat{y} \in \mathcal{Y}$ and $a \in \mathcal{A}$) such that the sum of the corresponding conditional probability density values stays the same. Generally speaking, if the way $f$ "picks out" $x$ is not strictly coupled with the conditional probability density $p_{X|AY}(x \mid a, y)$, the condition specified in Theorem 2 would be violated, i.e., Equalized Odds cannot hold true. In fact, one can formally establish the connection between Theorem 2 (for general regression cases) and Theorem 1 (for linear non-Gaussian regression cases) by specifying the way $f$ "picks out" $x$'s.[1] As we will see in Theorem 4, the phenomenon of unattainable Equalized Odds also occurs in classification with deterministic predictors.

### 3.3. Regression with Stochastic Prediction

In light of the unattainability of Equalized Odds for prediction with deterministic functions of $A$ and $X$, it is important to ask whether Equalized Odds can be attainable with stochastic prediction, where the output given input features is no longer a single value. As we shall see in Theorem 3, with stochastic prediction functions, under mild assumptions one can always find a non-trivial fair predictor $\widetilde{Y}$ that satisfies Equalized Odds, even if both the target $Y$ and the protected feature $A$ are continuous.

**Theorem 3** *(Attainability of Equalized Odds for Regression with Stochastic Prediction)*
*Let $A$, $X$, and $Y$ be continuous variables with domain of value $\mathcal{A}$, $\mathcal{X}$, and $\mathcal{Y}$, respectively. Assume that their joint distribution is fixed and known. Further assume $Y \not\perp\!\!\!\perp A$, $Y \not\perp\!\!\!\perp X$, and $Y \not\perp\!\!\!\perp X \mid A$. Without loss of generality let the conditional probability density $p_{X|AY}(x \mid a, y)$ be non-negative and finite. Then there exists $\widetilde{Y}$ with domain of value $\mathcal{Y}$ whose distribution is fully determined by $p_{\widetilde{Y}|AX}(\tilde{y} \mid a, x)$, such that $\widetilde{Y}$ is not independent from $(A, X)$ but $\widetilde{Y} \perp\!\!\!\perp A \mid Y$, i.e., the Equalized Odds is non-trivially attainable.*

From Theorem 3 we can see that with stochastic prediction, one can guarantee the attainability of Equalized Odds for regression under rather mild assumptions. It would then be desirable to construct general, nonlinear prediction models to produce a stochastic prediction (i.e., with a certain type of randomness in the prediction). One possible way follows the framework of Generative Adversarial Networks (GANs) (Goodfellow et al., 2014): we use random standard Gaussian noise $E$, in addition to $A$ and $X$, as input, such that the output will have a specific type of randomness. The parameters involved are learned by minimizing prediction error and enforcing Equalized Odds on the

---

1. The derivation of the connection between Theorem 2 and Theorem 1 can be found in Section C.1 of the appendix.

"randomized" output, as a function of $A$, $X$, and $E$, at the same time. Given the loss function $\mathcal{L}$, a class of function $\mathcal{F}$, and a fairness penalty $\mathcal{G}$, we propose the following objective function for fitting an Equalized Odds model with stochastic prediction:

$$\min_{f \in \mathcal{F}} \mathbb{E}\big[\mathcal{L}\big(f(A, X, E), Y\big) + \lambda \mathcal{G}(f(A, X, E), A, Y)\big]. \tag{5}$$

The first term measures the prediction error, for instance the mean squared error for regression. The second term is the fairness penalty that imposes Equalized Odds. We use the kernel measure of conditional dependence (Fukumizu et al., 2004, 2007) between $\widetilde{Y}$ and $A$ given $Y$ as the fairness penalty term. The hyperparameter $\lambda$ reflects the trade-off between accuracy and fairness.

## 4. Fairness in Classification

In this section, we consider the attainability of Equalized Odds for binary classifiers (with a deterministic or stochastic prediction function), and furthermore, if attainable, the optimality of performance of various computational procedures under the fairness criterion.

### 4.1. Classification with Deterministic Prediction

We begin with considering cases when classification is performed by a deterministic function of the input. Similar to the previous analysis for regression tasks with deterministic prediction functions in Section 3.2, we derive the conditions under which Equalized Odds can possibly hold true for classification with deterministic prediction functions.

**Theorem 4** *(Condition to Achieve Equalized Odds for Classification with Deterministic Prediction) Assume that the protected feature $A$ and $Y$ are dependent and that their joint probability $P(A, Y)$ (for discrete $A$) or joint probability density $p(A, Y)$ (for continuous $A$) is positive for every combination of possible values of $A$ and $Y$. Further assume that $Y$ is not fully determined by $A$, and that there are additional features $X$ that are not independent of $Y$. Let the output of the classifier $\widehat{Y}$ be a deterministic function $f : \mathcal{A} \times \mathcal{X} \to \mathcal{Y}$. Let $S_A^{(\hat{y})} := \{a \mid \exists x \in \mathcal{X} \text{ s.t. } f(a, x) = \hat{y}\}$, and $S_{X|a}^{(\hat{y})} := \{x \mid f(a, x) = \hat{y}\}$. Equalized Odds is attained if and only if the following conditions hold true (for continuous $X$, replace summation with integration accordingly):*

*(i) $\forall \hat{y} \in \mathcal{Y}: S_A^{(\hat{y})} = \mathcal{A}$,*

*(ii) $\forall \hat{y} \in \mathcal{Y}, \forall a, a' \in \mathcal{A}: \sum_{x \in S_{X|a}^{(\hat{y})}} P_{X|AY}(x \mid a, y) = \sum_{x \in S_{X|a'}^{(\hat{y})}} P_{X|AY}(x \mid a', y).$*

Condition (i) says that within each class determined by the classification function $f$, $A$ should be able to take all possible values in $\mathcal{A}$. While condition (i) is already rather restrictive, condition (ii) specifies an even stronger constraint on the relation between $P_{X|AY}(x|a, y)$ (or $p_{X|AY}(x|a, y)$ for continuous $X$) and the set $S_{X|a}^{(\hat{y})}$ (which is determined by the function $f$). Generally speaking, in order to score a better classification accuracy, one would like to make $P_{\widehat{Y}|A,X}(\hat{y}|a, x)$ as close as possible to $P_{Y|A,X}(y|a, x)$, and if the set $S_{X|a}^{(\hat{y})}$ and $P_{X|AY}(x|a, y)$ are not strictly coupled, condition (ii) would be violated, i.e., Equalized Odds cannot be attained.[2]

---

2. Interested readers may find in Section C of the appendix concrete examples where the aforementioned conditions can or cannot hold true.

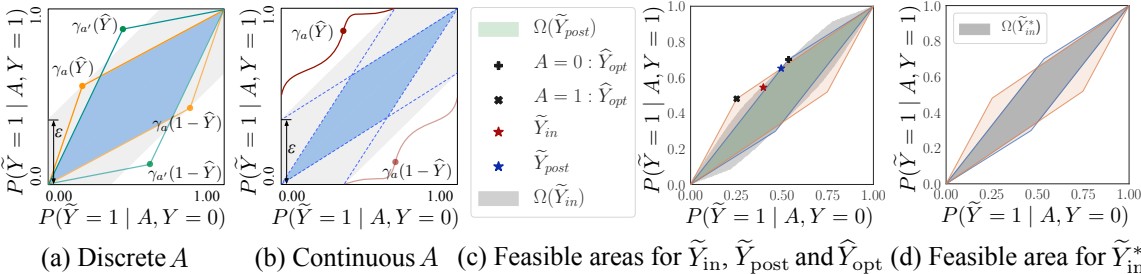

(a) Discrete $A$     (b) Continuous $A$    (c) Feasible areas for $\widetilde{Y}_{\text{in}}$, $\widetilde{Y}_{\text{post}}$ and $\widehat{Y}_{\text{opt}}$   (d) Feasible area for $\widetilde{Y}_{\text{in}}^{*}$

Figure 1: ROC feasible area illustrations. Panels (a)-(b): Attainability of Equalized Odds for binary classifiers with stochastic prediction functions. Panels (c)-(d): ROC feasible areas comparison between $\Omega(\widetilde{Y}_{\text{in}})$, $\Omega(\widetilde{Y}_{\text{post}})$, $\Omega(\widehat{Y}_{\text{opt}})$, and $\Omega(\widetilde{Y}_{\text{in}}^{*})$.

## 4.2. Classification with Stochastic Prediction

In this section, we consider cases when stochastic prediction is acceptable, namely, the classifier would output class labels with certain probabilities. Among different categories of approaches to derive a fair classifier, recent efforts to impose Equalized Odds in the pre-processing manner (Madras et al., 2018; Zhao et al., 2020; Tan et al., 2020) approach the problem from a representation learning perspective, where the main focus is to learn fair representations that at the same time preserve sufficient information from the original data. Therefore, we omit pre-processing approaches from the discussion and focus on post-processing and in-processing fair classifiers.

We first derive the relation between positive rates (TPR and FPR) of binary classifiers before and after the post-processing step, i.e., $\widehat{Y}_{\text{opt}}$ (the unconstrainedly optimized classifier) and $\widetilde{Y}_{\text{post}}$ (the fair classifier derived by post-processing $\widehat{Y}_{\text{opt}}$), and show that under mild assumptions, one can always derive a non-trivial Equalized Odds $\widetilde{Y}_{\text{post}}$ via a post-processing step. Then, from the ROC feasible area perspective, we prove that post-processing approaches are actually equivalent to in-processing approaches but with additional "pseudo" constraints enforced. Therefore, using the same loss function, post-processing approaches can perform no better than in-processing approaches.

### 4.2.1. THE POST-PROCESSING STEP

The post-processing step (Hardt et al., 2016) of a predictor $\widehat{Y}$ (here we drop the subscript when there is no ambiguity) only utilizes the information in the joint distribution $(A, Y, \widehat{Y})$. A fair predictor $\widetilde{Y}_{\text{post}}$ derived via a post-processing step is then fully specified by a (possibly randomized) function of $(A, \widehat{Y})$. This implies the conditional independence $\widetilde{Y}_{\text{post}} \perp\!\!\!\perp Y \mid A, \widehat{Y}$. Then if we denote the positive rates of $\widehat{Y}$ as $P_{\widehat{Y}|AY}(1|a, y)$, positive rates of $\widetilde{Y}_{\text{post}}$ as $P_{\widetilde{Y}_{\text{post}}|AY}(1|a, y)$, we can factorize the positive rates under this conditional independence: $P_{\widetilde{Y}_{\text{post}}|AY}(1|a, y) = \sum_{\hat{y} \in \mathcal{Y}} \beta_a^{(\hat{y})} P_{\widehat{Y}|AY}(\hat{y}|a, y)$, where $\beta_a^{(\hat{y})} := P(\widetilde{Y}_{\text{post}} = 1 \mid A = a, \widehat{Y} = \hat{y})$. Therefore, the post-processing step boils down to optimizing parameters (for discrete $A$) or functions (for continuous $A$) $\beta_a^{(\hat{y})}$.

## 4.2.2. ROC FEASIBLE AREA

On the Receiver Operator Characteristic (ROC) plane, a two-dimensional plane with horizontal axis denoting FPR and vertical axis denoting TPR, the performance of any binary predictor $\widehat{Y}$ (not necessarily a fair one) with a certain value of protected feature $A = a$ corresponds to a point $\gamma_a(\widehat{Y}) =$ (FPR, TPR) on the plane. Denote each coordinate according to the value of $Y$ as $\gamma_{ay}(\widehat{Y})$:

$$\gamma_a(\widehat{Y}) = \big(\gamma_{a0}(\widehat{Y}), \gamma_{a1}(\widehat{Y})\big) := \big(P_{\widehat{Y}|AY}(1|a,0), P_{\widehat{Y}|AY}(1|a,1)\big). \tag{6}$$

Further denote the corresponding convex hull of $\widehat{Y}$ on the ROC plane as $\mathcal{C}_a(\widehat{Y})$ using vertices:

$$\mathcal{C}_a(\widehat{Y}) := \text{Conv}\big\{(0,0), \gamma_a(\widehat{Y}), \gamma_a(1-\widehat{Y}), (1,1)\big\}, \tag{7}$$

and then, as already stated in Hardt et al. (2016), the (FPR, TPR) pair corresponding to a post-processing predictor falls within (including the boundary of) $\mathcal{C}_a(\widehat{Y})$.

**Definition 5 (ROC feasible area)** *The feasible area of a predictor $\Omega(\widehat{Y})$, specified by the hypothesis space of available predictors $\widehat{Y}$, is the set containing all attainable (FPR, TPR) pairs by the predictor on the ROC plane satisfying Equalized Odds.*

Following Hardt et al. (2016), we analyze the relation between the (FPR, TPR) pair of predictors on the ROC plane and formally establish the existence of the non-trivial Equalized Odds predictor. As we show in Theorem 6, under mild assumptions an Equalized Odds predictor $\widetilde{Y}_{\text{post}}$ derived via post-processing $\widehat{Y}$ always has non-empty ROC feasible area.

**Theorem 6** *(Attainability of Equalized Odds for Classification with Stochastic Prediction)* *Assume that the feature $X$ is not independent from $Y$, and that $\widehat{Y}$ is a function of $A$ and $X$. Then for binary classification, if $\widehat{Y}$ is a non-trivial predictor for $Y$, there is always at least one non-trivial predictor $\widetilde{Y}_{post}$ derived by post-processing $\widehat{Y}$ that can attain Equalized Odds, i.e., $\Omega(\widetilde{Y}_{post}) \neq \emptyset$.*

Here $\widetilde{Y}_{\text{post}}$ is a possibly randomized function of only $A$ and $\widehat{Y}$, trading off TPR with FPR across groups with different value of protected feature. From the panels (a) and (b) of Figure 1 we can also see that $\Omega(\widetilde{Y}_{\text{post}})$, the ROC feasible area of $\widetilde{Y}_{\text{post}}$, is the intersection of $\Omega_a(\widehat{Y})$, indicating that although Equalized Odds is attained, the performance of $\widetilde{Y}_{\text{post}}$ is no better and often worse than the weakest performance across different groups, which is obviously suboptimal.

## 4.2.3. OPTIMALITY OF PERFORMANCE AMONG FAIR CLASSIFIERS

In this subsection we discuss the optimality of performance of fair classifiers derived via in-processing and post-processing approaches. Let us consider a more general setting for the post-processing approach, where the optimal predictor to be post-processed is stochastic (while most previous approaches use deterministic predictors). We can derive the in-processing fair predictor $\widetilde{Y}_{\text{in}}$:

$$\begin{aligned} \min_{f \in \mathcal{F}} \quad & \mathbb{E}[\mathcal{L}(\widetilde{Y}_{\text{in}}, Y)] \\ \text{s.t.} \quad & P_{\widetilde{Y}_{\text{in}}|AY}(\tilde{y} \mid a, y) = P_{\widetilde{Y}_{\text{in}}|Y}(\tilde{y} \mid y) \\ \text{where} \quad & \widetilde{Y}_{\text{in}} \sim \text{Bernoulli}\big(f(A, X)\big); \end{aligned} \tag{8}$$

and the unconstrained statistical optimal predictor $\widehat{Y}_{\text{opt}}$:

$$\min_{f \in \mathcal{F}} \quad \mathbb{E}[\mathcal{L}(\widehat{Y}_{\text{opt}}, Y)]$$
$$\text{where} \quad \widehat{Y}_{\text{opt}} \sim \text{Bernoulli}\big(f(A, X)\big). \tag{9}$$

It is natural to wonder, now that one can always directly solve for $\widetilde{Y}_{\text{in}}$ from Equation 8, how it is related to $\widetilde{Y}_{\text{post}}$, which is derived by post-processing the $\widehat{Y}_{\text{opt}}$ solved from Equation 9? Interestingly, although $\widetilde{Y}_{\text{in}}$ and $\widetilde{Y}_{\text{post}}$ are solved separately using different constrained optimization schemes, one can draw a connection between them by utilizing $\widehat{Y}_{\text{opt}}$ as a bridge and reason about the relation between their ROC feasible areas $\Omega(\widetilde{Y}_{\text{in}})$ and $\Omega(\widetilde{Y}_{\text{post}})$, as we summarize in the following theorem.

**Theorem 7** *(**Equivalence between ROC feasible areas**)*
*Let $\Omega(\widetilde{Y}_{post})$ denote the ROC feasible area specified by the constraints enforced on $\widetilde{Y}_{post}$. Then $\Omega(\widetilde{Y}_{post})$ is identical to the ROC feasible area $\Omega(\widetilde{Y}_{in}^*)$ that is specified by the following set of constraints:*
  *(i) constraints enforced on $\widetilde{Y}_{in}$ ;*
  *(ii) additional "pseudo" constraints: $\forall a \in \mathcal{A}$, $\beta_{a0}^{(0)} = \beta_{a1}^{(0)}$, $\beta_{a0}^{(1)} = \beta_{a1}^{(1)}$, where*
    $\beta_{ay}^{(\hat{y})} = \sum_{x \in \mathcal{X}} P(\widetilde{Y}_{in} = 1 \mid A = a, X = x) P(X = x \mid A = a, Y = y, \widehat{Y}_{opt} = \hat{y})$.

As we can see in Figure 1(c) and (d), if the additional "pseudo" constraints are introduced when optimizing $\widetilde{Y}_{\text{in}}^*$, we have $\Omega(\widetilde{Y}_{\text{in}}) \supseteq \Omega(\widetilde{Y}_{\text{post}}) = \Omega(\widetilde{Y}_{\text{in}}^*)$. Therefore, with the same objective function and fairness constraint, the fair classifier derived from an in-processing approach always outperforms (or performs equally well with) the one derived from a post-processing approach.

## 5. Experiments

In this section, we provide numerical results for various settings. To demonstrate the benefit of stochastic prediction with respect to imposing fairness, we present the results for regression with stochastic prediction on both simulated data and the real-world *Communities and Crime* data set. For classification tasks we compare the performance of several existing methods in the literature on multiple real-world data sets. Please see Section B of the appendix for additional results, as well as the detailed description of the data sets and technical specification for the experiments.

In Figure 2 we compare deterministic and stochastic regression on simulated linear non-Gaussian data. Panel (a) illustrates the trade-off between fairness in terms of the kernel measure of conditional dependence (KMCD) (Fukumizu et al., 2004, 2007), and prediction error in terms of the mean squared error (MSE) for different values of the hyperparameter $\lambda$. Panel (b) summarizes the p-value outputs of the kernel-based conditional independence (KCI) test (Zhang et al., 2011). Note that in (b), the green boxes are not visible since p-values in the deterministic prediction case are always close to 0. The distribution of the p-value outputs clearly indicate that stochastic prediction often achieve Equalized Odds (the test fails to reject the null quite often), while deterministic prediction may not (the test rejects the null almost all the time). For panel (c) and (d), we fix the training sample size at 200 while the sample size for testing ranges from 100 to 3000.[3] A practical example of the

---

3. Considering the fact that the computational cost for KCI test p-value increases dramatically with large sample size, we use linearly correlated data with relatively small sample size for training and consider increasingly large test sets in our repetitive experiments (with fixed hyperparameter $\lambda$).

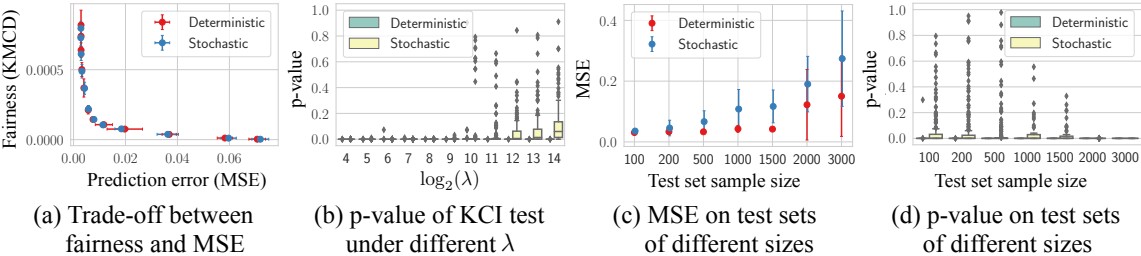

Figure 2: Illustration of the benefit of the stochastic prediction for the purpose of imposing Equalized Odds. The data is linear non-Gaussian, and the predictor is a neural network (nonlinear) regressor with deterministic or stochastic output. The green boxes in panel (b) and (d) are barely visible, meaning that p-values for deterministic prediction are always close to $0$.

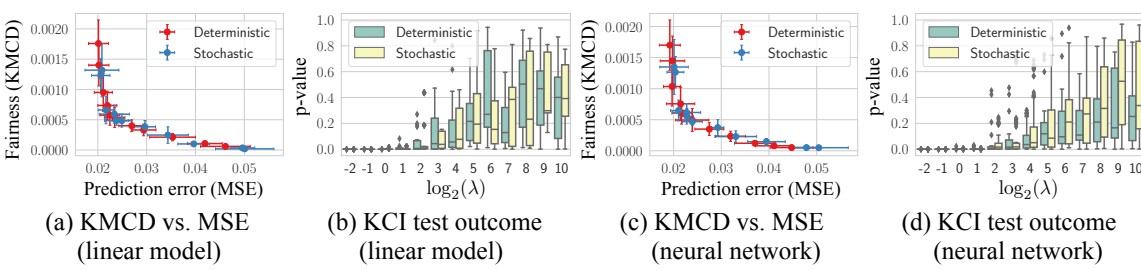

Figure 3: Results for regression with deterministic and stochastic prediction functions on the *Communities and Crime* data set. The base model could be a linear regressor or a neural network regressor, and the stochastic prediction is achieved by introducing an independent noise input sampled from a standard Gaussian distribution.

scenario would be an automated hiring system to aid recruitment (each future applicant is a new data point for the system as long as the system is deployed and keeps running). As we can see in panel (d), while the p-value for deterministic prediction is always close to $0$ (invisible green box, which indicates violated Equalized Odds), the p-value for stochastic prediction spreads over $[0, 1]$ if the test set sample size is not too big compared to the training data.

In Figure 3 we compare deterministic and stochastic regression on the *Communities and Crime* data set. Compared to Figure 2(b), although the green boxes (which correspond to the distribution of p-values for the deterministic predictor) are visible in Figure 3(d), we can still observe that the distribution of p-values for the stochastic predictor dominates its counterpart for deterministic predictor, indicating a higher possibility of attaining Equalized Odds. Compare the KCI test outcome of the linear stochastic model shown in panel (b) with that of the nonlinear stochastic model (implemented with the neural network) shown in panel (d), we can see that the distribution of p-values in panel (b) for stochastic prediction (with a linear base model) does not dominate its deterministic counterpart. This is not surprising since if the deterministic part of the linear model does not satisfy Equalized Odds, adding an independent noise will not help impose fairness.

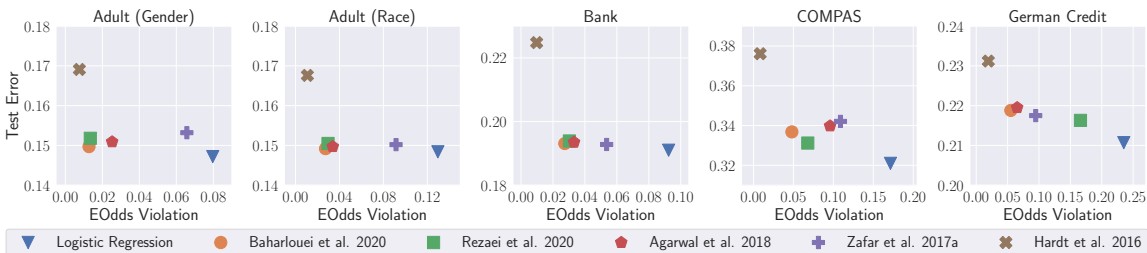

Figure 4: Experimental results for classification with Equalized Odds criterion.

In Figure 4, we compare the performance under Equalized Odds of multiple methods proposed in the literature. Although a probabilistic classification model is used for across each method (logistic regression here), if an algorithm outputs the class label where the prediction likelihood is maximized, the prediction is in essence deterministic. Therefore, although here we are considering finite data cases, we can still anticipate a lower level of fairness violation with stochastic prediction. This is validated by the numerical experiment: while the post-processing approach by Hardt et al. (2016) does not score the lowest test error, the violation of Equalized Odds is the lowest compared to other approaches.

## 6. Further Discussions

In this section, we first present an alternative way to perform post-processing for binary classification, and discuss how our results would apply to this specific post-processing strategy; we then discuss the relation between our findings and previous theoretical results for Equalized Odds in the literature.

### 6.1. Alternative Strategy for Post-processing

When we discuss Equalized Odds fairness in binary classification, we present the attainability and optimality results by considering one particular type of post-processing approach proposed by Hardt et al. (2016), where $\widetilde{Y}_{\text{post}}$ is derived by first training $\widehat{Y}$ without fairness constraints and then adjusting the output in a post-hoc way (also referred to as the "two-step framework" in Woodworth et al. (2017)). In the literature, the term "post-processing" is also used to describe the prediction mechanism where the classification result is derived from thresholding a non-dichotomized predictor, e.g., the Bayes optimal regressor $R$ (Hardt et al., 2016).

If the classifier $\widetilde{Y}_{\text{post}}$ is derived from post-processing the non-dichotomized $R$ instead of the binary $\widehat{Y}$, in general the conditional independence $\widetilde{Y}_{\text{post}} \perp\!\!\!\perp Y \mid A, \widehat{Y}$ no longer holds true since the non-dichotomized $R$ contains more information about $Y$ than $\widehat{Y}$. That being said, the attainability of Equalized Odds for post-processing is still guaranteed, in other words, the result of Theorem 6 holds true even for this alternative post-processing strategy. We can construct a proof similar to that for Theorem 3 (the attainability result for regression tasks) but with a discrete domain $\mathcal{Y}$. Specifically, the modifications of the proof include (1) changing integrations with respect to $\mathcal{Y}$ to summations, and (2) replacing conditional densities $p_{\widetilde{Y}|AX}(\tilde{y} \mid a, x)$ with the corresponding conditional probabilities $P_{\widetilde{Y}|AX}(\tilde{y} \mid a, x)$. Besides, if we post-process the Bayes optimal regressor $R$ to derive $\widetilde{Y}_{\text{post}}$ for binary classification, the optimality of performance can also be guaranteed. It is shown by Hardt et al. (2016, Proposition 5.2) that the Bayes optimal Equalized Odds binary classifier $\widetilde{Y}_{\text{post}}$ can be derived by

post-processing the Bayes optimal regressor $R$. The optimality of such classifiers does not contradict our result based on post-processing $\widehat{Y}$ and the results can be viewed as complementary to each other.

## 6.2. Discussion on Previous Theoretical Results for Equalized Odds in Binary Classification

Apart from various methods to empirically impose Equalized Odds (Hardt et al., 2016; Woodworth et al., 2017; Agarwal et al., 2018; Mary et al., 2019; Baharlouei et al., 2020; Rezaei et al., 2020; Romano et al., 2020), there are also theoretical considerations with respect to this specific notion of fairness. In particular, for binary classification tasks, Hardt et al. (2016, Proposition 5.2) show that if we have access to the Bayes optimal regressor $R$, we can derive the Bayes optimal Equalized Odds binary classifier by post-processing $R$; Woodworth et al. (2017, Theorem 7) show that for a binary classifier hypothesis class, one can derive a guarantee regarding the statistical learnability of (if there exists) the best Equalized Odds predictor within the class using the two-step framework.

Previous theoretical results for Equalized Odds require strong assumptions: the result by Hardt et al. (2016, Proposition 5.2) requires the availability of the Bayes optimal regressor $R$; the concentration bounds (on both accuracy and fairness violation) demonstrated by Woodworth et al. (2017, Theorem 7) assume the existence of the best non-discriminatory binary classifier within the hypothesis class. Instead of assuming the availability of Bayes optimal regressor (as compared to Hardt et al. (2016, Proposition 5.2)) or the existence of a perfect fair predictor (as compared to Woodworth et al. (2017, Theorem 7)), we give an affirmative answer to the necessary and sufficient condition under which Equalized Odds can possibly hold true (for deterministic predictors) and also a guarantee on the attainability of Equalized Odds under rather mild assumptions (for stochastic predictors).

Our theoretical analysis regarding the attainability of Equalized Odds is actually the missing piece in the puzzle, which, together with previous theoretical results by Hardt et al. (2016, Proposition 5.2) and Woodworth et al. (2017, Theorem 7), gives us a much clearer picture regarding achieving Equalized Odds for binary classification: the stochastic prediction scheme is preferable since the theoretical attainability guarantee for the perfect Equalized Odds in the large sample limit; one can also characterize the price paid in terms of accuracy when using the stochastic prediction scheme by analyzing the bound for classification error and that for empirical fairness violation. Furthermore, our attainability results apply not only to binary classification tasks, but also to regression tasks.[4]

## 6.3. The Difficulty of Deriving Bounds for Accuracy and Fairness Violation in General Cases

One limitation of our work is that we only consider the attainability of the perfect fairness and does not provide a lower bound of best achievable fairness violation when attainability is not guaranteed. However, as illustrated in Theorem 2 and Theorem 4 (the necessary and sufficient conditions for Equalized Odds to hold true with the deterministic prediction scheme), the violation of Equalized Odds for deterministic predictors directly corresponds to the specific properties of the data distribution itself. This indicates the difficulty of giving a tight universal distribution-free lower bound for fairness violation and accuracy in the large sample limit for deterministic predictors. As a comparison, for the stochastic prediction scheme our results (Theorem 3 and Theorem 6) show that in general we have theoretical attainability guarantees for Equalized Odds, i.e., a distribution-free lower bound (at zero)

---

4. With modifications to the proof of Theorem 3 (as discussed in Section 6.1), one can actually also construct a proof and apply our attainability results to multiclass classification tasks.

of fairness violation. Only with this theoretical guanrantee can we apply the result by Woodworth et al. (2017, Theorem 7) and analyze concentration bounds for binary classification.[5]

In fact, the difficulty of theoretically deriving (distribution-free or distribution-dependent) bounds in general cases, especially for tasks other than binary classification, stems from the following aspects: the quantification of fairness violation, and the exact mathematical form of the trade-off between accuracy and fairness. To begin with, the quantification of fairness violation could be involved beyond binary classification. For binary classification tasks, we can conveniently quantify Equalized Odds violation via TPR(s) and FPR(s) that directly link back to distribution specifics (e.g., the joint distribution of $(A, Y)$), but this is not the case for multiclass classification or regression tasks. A natural choice of fairness violation quantification beyond binary classification is, for example, the (normalized) kernel measure of conditional dependence (KMCD). However, it is not intuitive how to effectively find a preimage of the KMCD operator and translate the fairness violation in terms of distributional properties of the data. Furthermore, although we can empirically draw trade-off curves, it is not transparent what would be the explicit mathematical expression of the trade-off (in the large sample limit) between accuracy and fairness violation. The explicit expression of the trade-off is essential if we were to, for example, compare the (empirically) best possible deterministic predictor with its stochastic counterpart and derive a theoretical bound.

### 6.4. Potential Social Impact of Stochastic Prediction

Under the scope of the broader social impact, the stochastic prediction to impose a group-level fairness like Equalized Odds may subject to unwanted usage, which may result in deterministic (therefore in general unfair) prediction in the end.[6] The introduction of randomness in the prediction may also be subject to concerns, especially in high-stake decision-makings. In some way, this potentially negative effect roots from the fact that Equalized Odds is a group-level fairness criterion, i.e., not fine-grained enough to characterize individual-level consequences. In high-stakes scenarios when the decisions received by individuals matter, we should dedicate to more fine-grained fairness notions which go beyond group-level characterization of non-discrimination.

## 7. Conclusion

In this paper, we focus on Equalized Odds and consider the attainability of fairness, and furthermore, if it is attainable, the optimality of the prediction performance under various settings. For both classification and regression tasks, we present necessary and sufficient conditions under which Equalized Odds can hold true with deterministic prediction functions; we also present theoretical guarantees that under mild assumptions, one can always find a non-trivial stochastic predictor that satisfies Equalized Odds in the large sample limit, while in general fairness is not attainable with a deterministic predictor.

We hope that our results can incentivize further research into developing prediction strategies that come with theoretical guarantees. Future work would naturally consider the attainability of more fine-grained (compared to group fairness) criteria of fairness (e.g., individual fairness) as well as fairness notions that focus on discrimination involved in the data generating process.

---

5. Otherwise, the assumption of the result by Woodworth et al. (2017, Theorem 7) that there exist such a non-trivial fair predictor in the hypothesis class might be violated. For example, if the hypothesis class contains only deterministic predictors, our results (Theorem 2 and Theorem 4) indicate that this assumption is in general violated.

6. Interested readers may find a practical example in Section D of the appendix.

## Acknowledgments

KZ would like to acknowledge the support by the National Institutes of Health (NIH) under Contract R01HL159805, by the NSF-Convergence Accelerator Track-D award #2134901, and by the United States Air Force under Contract No. FA8650-17-C7715.

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

## Appendix A. Proof for Theorems

### A.1. Proof for Theorem 1

To prove the unattainability of Equalized Odds in regression, we will need the following lemma, which provides a way to characterize conditional independence/dependence with conditional or joint distributions.

**Lemma 8** *Variables $V_1$ and $V_2$ are conditionally independent given variable $V_3$ if and only if there exist functions $h(v_1, v_3)$ and $g(v_2, v_3)$ such that*

$$p_{V_1, V_2 | V_3}(v_1, v_2 \mid v_3) = h(v_1, v_3) \cdot g(v_2, v_3). \tag{10}$$

**Proof** First, if $V_1$ and $V_2$ are conditionally independent given variable $V_3$, then Equation 10 holds:

$$p_{V_1, V_2 | V_3}(v_1, v_2 \mid v_3) = p_{V_1 | V_3}(v_1 \mid v_3) \cdot p_{V_2 | V_3}(v_2 \mid v_3).$$

We then let $\tilde{h}(v_3) := \int h(v_1, v_3) dv_1$ and $\tilde{g}(v_3) := \int g(v_2, v_3) dv_2$. Take the integral of Equation 10 w.r.t. $v_1$ and $v_2$, we have:

$$p_{V_2 | V_3}(v_2 \mid v_3) = \tilde{h}(v_3) \cdot g(v_2, v_3),$$
$$p_{V_1 | V_3}(v_1 \mid v_3) = \tilde{g}(v_3) \cdot h(v_1, v_3),$$

respectively. Bearing in mind Equation 10, one can see that the product of the two equations above is

$$p_{V_2 | V_3}(v_2 \mid v_3) \cdot p_{V_1 | V_3}(v_1 \mid v_3)$$
$$= \tilde{h}(v_3) \cdot g(v_2, v_3) \cdot \tilde{g}(v_3) \cdot h(v_2, v_3)$$
$$= \tilde{h}(v_3) \cdot \tilde{g}(v_3) \cdot p_{V_1, V_2 | V_3}(v_1, v_2 \mid v_3).$$

Take the integral of the equation above w.r.t. $v_1$ and $v_2$ gives $\tilde{h}(v_3) \cdot \tilde{g}(v_3) \equiv 1$. The above equation then reduces to

$$p_{V_2 | V_3}(v_2 \mid v_3) \cdot p_{V_1 | V_3}(v_1 \mid v_3) = p_{V_1, V_2 | V_3}(v_1, v_2 \mid v_3).$$

That is, $V_1$ and $V_2$ are conditionally independent given $V_3$. ∎

Now we are ready to prove the unattainability of Equalized Odds in linear non-Gaussian regression. Recall that the data is generated as follows ($H$ is not measured in the dataset):

$$X = qA + E_X,$$
$$H = bA + E_H, \tag{11}$$
$$Y = cX + dH + E_Y,$$

where $(A, E_X, E_H, E_Y)$ are mutually independent and $q, b, c, d$ are constants.

**Theorem** *(Unattainability of Equalized Odds in the Linear Non-Gaussian Case)*
*Assume that $X$ has a causal influence on $Y$, i.e., $c \neq 0$ in Equation 11, and that $A$ and $Y$ are not independent, i.e., $qc + bd \neq 0$. Assume $p_{E_X}$ and $p_E$ are positive on $\mathbb{R}$. Let $f_1 := \log p_A$, $f_2 := \log p_{E_X}$, and $f_3 := \log p_E$. Further assume that $f_2$ and $f_3$ are third-order differentiable. Then if at most one of $E_X$ and $E$ is Gaussian, $\widehat{Y}$ is always conditionally dependent on $A$ given $Y$.*

**Proof** According to Equation 11, we have

$$
\begin{bmatrix} A \\ \widehat{Y} \\ Y \end{bmatrix} = \begin{bmatrix} 1 & 0 & 0 \\ \alpha + q\beta & \beta & 0 \\ qc + bd & c & 1 \end{bmatrix} \cdot \begin{bmatrix} A \\ E_X \\ E \end{bmatrix}. \tag{12}
$$

The determinant of the above linear transformation is $\beta$, which relates the probability density function of the variables on the LHS and that of the variables on the RHS of the equation. Therefore, according to Equation 12, we can rewrite the joint probability density function by making use of the Jacobian determinant and factor the joint density into marginal density functions ($A$, $E_X$, $E$ are mutually independent according to the data generating process). Further let

$$
\tilde{\alpha} := \frac{\alpha + q\beta}{\beta}, \quad \tilde{r} := bd - \frac{c\alpha}{\beta}, \quad \text{and } \tilde{c} := \frac{c}{\beta}. \tag{13}
$$

Then we have $E_X = \frac{1}{\beta}\widehat{Y} - \tilde{\alpha}A$, $E = Y - \tilde{r}A - \tilde{c}\widehat{Y}$, and

$$
\begin{aligned}
& p_{A,\widehat{Y},Y}(a, \hat{y}, y) \\
&= p_{A,E_X,E}(a, e_x, e)/|\beta| \\
&= \frac{1}{|\beta|}p_A(a)p_{E_X}(e_x)p_E(e) \\
&= \frac{1}{|\beta|}p_A(a)p_{E_X}(\frac{1}{\beta}t - \tilde{\alpha}a)p_E(y - \tilde{r}a - \tilde{c}\hat{y}).
\end{aligned}
$$

On its support, the log-density can be written as

$$
\begin{aligned}
J &:= \log p_{A,\widehat{Y},Y}(a, \hat{y}, y) \\
&= \log p_A(a) + \log p_{E_X}(\frac{1}{\beta}\hat{y} - \tilde{\alpha}a) \\
&\quad + \log p_E(y - \tilde{r}a - \tilde{c}\hat{y}) - log|\beta| \\
&= f_1(a) + f_2(\frac{1}{\beta}\hat{y} - \tilde{\alpha}a) \\
&\quad + f_3(y - \tilde{r}a - \tilde{c}\hat{y}) - \log|\beta|.
\end{aligned} \tag{14}
$$

According to Lemma 8, $A \perp\!\!\!\perp \widehat{Y} \mid Y$ if and only if $p_{A,\widehat{Y}|Y}(a, \hat{y} \mid y)$ is a product of a function of $a$ and $y$ and a function of $\hat{y}$ and $y$. $p_{A,\widehat{Y},Y}(a, \hat{y}, y)$ is further a product of the above function and a function of only $y$. This property, under the conditions in Theorem 1, is equivalent to the constraint

$$
\frac{\partial^2 J}{\partial A \partial \widehat{Y}} \equiv 0. \tag{15}
$$

According to Equation 14, we have

$$
\frac{\partial J}{\partial \hat{y}} = \frac{1}{\beta} \cdot f_2'(\frac{1}{\beta}\hat{y} - \tilde{\alpha}a) - \tilde{c} \cdot f_3'(y - \tilde{r}a - \tilde{c}\hat{y}),
$$

and therefore

$$
\frac{\partial^2 J}{\partial a \partial \hat{y}} = -\frac{\tilde{\alpha}}{\beta} \cdot f_2''(\frac{1}{\beta}\hat{y} - \tilde{\alpha}a) + \tilde{r}\tilde{c} \cdot f_3''(y - \tilde{r}a - \tilde{c}\hat{y}). \tag{16}
$$

Combining Equations 15 and 16 gives

$$\tilde{r}\tilde{c} \cdot f_3''(y - \tilde{r}a - \tilde{c}\hat{y}) = \frac{\tilde{\alpha}}{\beta} \cdot f_2''(\frac{1}{\beta}\hat{y} - \tilde{\alpha}a). \tag{17}$$

Further taking the partial derivative of both sides of the above equation w.r.t. $y$ yields

$$\tilde{r}\tilde{c} \cdot f_3'''(y - \tilde{r}a - \tilde{c}\hat{y}) \equiv 0. \tag{18}$$

There are three possible situations where the above equation holds:

(i) $\tilde{c} = 0$, which is equivalent to $c = 0$ and contradicts with the theorem assumption.

(ii) $\tilde{r} = 0$. Then according to Equation 17, we have $\frac{\tilde{\alpha}}{\beta} \cdot f_2''(\frac{1}{\beta}\hat{y} - \tilde{\alpha}a) \equiv 0$, implies either $\tilde{\alpha} = 0$ or $f_2''(\frac{1}{\beta}\hat{y} - \tilde{\alpha}a) \equiv 0$. If the latter is the case, then $f_2$ is a linear function and, accordingly, $\exp(f_2)$ is not integrable and does not correspond to any valid density function. If the former is true, i.e., $\tilde{\alpha} = 0$, then according to Equation 13, we have $\alpha = -q\beta$, which further implies $\tilde{r} = bd - \frac{c\alpha}{\beta} = bd + qc$. Therefore, in this situation, $bd + qc = 0$, which again contradicts with the theorem assumption.

(iii) $f_3'''(y - \tilde{r}a - \tilde{c}\hat{y}) \equiv 0$. That is, $f_3$ is a quadratic function with a nonzero coefficient for the quadratic term (otherwise $f_3$ does not correspond to the logarithm of any valid density function). Thus $E$ follows a Gaussian distribution.

Only situation (iii) is possible, i.e., $\tilde{r}\tilde{c} \neq 0$ and $E$ follows a Gaussian distribution. This further tells us that the RHS of Equation 17 is a nonzero constant. Hence $f_2$ is a quadratic function and $E_X$ also follows a Gaussian distribution. Therefore if $A \perp\!\!\!\perp \hat{Y} \mid Y$ were to be true, then $E_X$ and $E$ are both Gaussian. Its contrapositive gives the conclusion of this theorem. ∎

**Corollary 9** *Suppose that both $E_X$ and $E$ are Gaussian, with variances $\sigma_{E_X}^2$ and $\sigma_E^2$, respectively. (The protected feature $A$ is not necessarily Gaussian.) Then $\hat{Y} \perp\!\!\!\perp A \mid Y$ if and only if*

$$\frac{\alpha}{\beta} = \frac{bdc \cdot \sigma_{E_X}^2 - q \cdot \sigma_E^2}{c^2 \cdot \sigma_{E_X}^2 + \sigma_E^2}. \tag{19}$$

**Proof** Under the condition that $E_X$ and $E$ are Gaussian, their log-density functions are third-order differentiable. Then according to the proof of Theorem 1, the Equalized Odds condition $A \perp\!\!\!\perp \hat{Y} \mid Y$ is equivalent to Equation 17, which, together with Equation 13 as well as the fact that $f_2'' = -\frac{1}{\sigma_{E_X}^2}$ and $f_3'' = -\frac{1}{\sigma_E^2}$, yields Equation 19. ∎

### A.2. Proof for Theorem 2

**Theorem** *(Condition to Achieve Equalized Odds for Regression with Deterministic Prediction)* *Assume that the protected feature $A$ and the continuous target variable $Y$ are dependent and that their joint probability density $p(A, Y)$ is positive for every combination of possible values of $A$ and $Y$. Further assume that $Y$ is not fully determined by $A$, and that there are additional features $X$ that are not independent of $Y$. Let the prediction $\hat{Y}$ be characterized by a deterministic function*

$f : \mathcal{A} \times \mathcal{X} \to \mathcal{Y}$. *Equalized Odds holds true if and only if the following condition is satisfied ($\delta(\cdot)$ is the delta function):*

$$\forall y, \hat{y} \in \mathcal{Y}, \ \forall a, a' \in \mathcal{A}, a \neq a' : \ Q(a, y, \hat{y}) = Q(a', y, \hat{y}),$$

$$\text{where } \ Q(a, y, \hat{y}) \triangleq \int_{\mathcal{X}} \delta(\hat{y} - f(a, x)) p_{X|AY}(x|a, y) dx.$$

**Proof** Without loss of generality, let us assume that $A$ and $X$ are continuous. By Lemma 8, the Equalized Odds criterion can be written into terms of the conditional probability density functions:

$$\forall y, \hat{y} \in \mathcal{Y}, \ a \in \mathcal{A} : \ p_{\widehat{Y}|AY}(\hat{y}|a, y) = p_{\widehat{Y}|Y}(\hat{y}|y) \tag{20}$$

Since $\widehat{Y} = f(A, X)$ where $f$ is a deterministic function of $(A, X)$, $f$ is an injective mapping from $\mathcal{A} \times \mathcal{X}$ to $\mathcal{Y}$. One can derive the joint probability density of $(\widehat{Y}, A, X, Y)$ by making use of the change of variables. Define the mapping $\mathcal{H}_f$ as following:

$$\mathcal{H}_f \begin{pmatrix} V \\ A \\ X \\ Y \end{pmatrix} = \begin{pmatrix} V + f(A, X) \\ A \\ X \\ Y \end{pmatrix},$$

where $V$ is a constant 0 whose probability density function is $\delta(v)$. Notice that $\mathcal{H}$ is bijective:

$$\mathcal{H}_f^{-1} \begin{pmatrix} \widehat{Y} \\ A \\ X \\ Y \end{pmatrix} = \begin{pmatrix} \widehat{Y} - f(A, X) \\ A \\ X \\ Y \end{pmatrix},$$

with the Jacobian matrix

$$J(\mathcal{H}_f^{-1}) = \begin{bmatrix} 1 & -\frac{\partial f}{\partial A} & -\frac{\partial f}{\partial X} & 0 \\ 0 & 1 & 0 & 0 \\ 0 & 0 & 1 & 0 \\ 0 & 0 & 0 & 1 \end{bmatrix}.$$

Therefore by making use of the change of variable technique, we have:

$$p_{\widehat{Y}AXY}(\hat{y}, a, x, y) = p_{VAXY}(v, a, x, y)|J(\mathcal{H}_f^{-1})|$$
$$= \delta(\hat{y} - f(a, x)) p_{AXY}(a, x, y).$$

Expand the LHS of Equation 20, we have:

$$p_{\widehat{Y}|AY}(\hat{y}|a, y) = \int_{\mathcal{X}} p_{\widehat{Y}X|AY}(\hat{y}, x|a, y) dx$$
$$= \int_{\mathcal{X}} \delta(\hat{y} - f(a, x)) p_{X|AY}(x|a, y) dx$$
$$:= Q(a, y, \hat{y}).$$

Since Equalized Odds holds true if and only if Equation 20 holds true, the LHS of the equation does not involve $a$ (as in RHS of the equation), i.e., $Q(a, y, \hat{y})$ does not change with $a$. Then, rewrite the RHS of Equation 20, we have:

$$
\begin{aligned}
p_{\widehat{Y}|Y}(\hat{y}|y) &= \int_{\mathcal{A}} p_{\widehat{Y}|AY}(\hat{y}|a, y) p_{A|Y}(a|y) da \\
&= \int_{\mathcal{A}} Q(a, y, \hat{y}) p_{A|Y}(a|y) da \\
&= Q(a, y, \hat{y}) \int_{\mathcal{A}} p_{A|Y}(a|y) da \\
&= Q(a, y, \hat{y}).
\end{aligned}
$$

Therefore, Equalized Odds implies the condition that $Q(a, y, \hat{y}) = Q(a', y, \hat{y})$. On the other hand, it is easy to see that when the aforementioned condition holds true, Equation 20 holds true, i.e., Equalized Odds holds true. ∎

### A.3. Proof for Theorem 3

In order to prove Theorem 3, we need the following results from functional analysis (a fuller treatment can be found, for example, in Meise and Vogt (1997); Daners (2008)):

**Lemma 10** *Let $\mathcal{S}$ be a set and $\mathcal{E} = (\mathcal{E}, \|\cdot\|)$ a Banach space. For a function $f : \mathcal{S} \to \mathcal{E}$, we define the supremum norm by $\|f\|_{\infty} := \sup_{s \in \mathcal{S}} \|f(s)\|$, and the space of bounded functions by $B(\mathcal{S}, \mathcal{E}) := \{f : \mathcal{S} \to \mathcal{E} \mid \|f\|_{\infty} < \infty\}$. Then $B(\mathcal{S}, \mathcal{E})$ is a Banach space with the supremum norm.*

**Definition 11 (Extreme Point)** *If $\mathcal{K}$ is a non-empty convex subset of a vector space, a point $x \in \mathcal{K}$ is called an extreme point of $\mathcal{K}$ if whenever $x = \lambda x_1 + (1 - \lambda)x_2$ with $0 < \lambda < 1$ and $x_1, x_2 \in \mathcal{K}$, then $x = x_1 = x_2$.*

**Theorem 12 (Krein-Milman Theorem (Krein and Milman, 1940))** *Let $\mathcal{K}$ be a non-empty compact subset of a locally convex Hausdorff topological vector space. Then the set of extreme points of $\mathcal{K}$ is not empty. If $\mathcal{K}$ is also convex, then $\mathcal{K}$ is the closed convex hull of its extreme points.*

It is a known result in finite-dimensional linear programming (see, for example, Dantzig (1965)) that if the specified feasible region $\mathcal{K}$ (which is the intersection of a finite number of half spaces) is non-empty, then the set of extreme points is not empty and finite, and the set of extreme directions is empty if and only if $\mathcal{K}$ is bounded. If $\mathcal{K}$ is unbounded, the set of extreme directions is not empty and finite. Furthermore, a point is in $\mathcal{K}$ if and only if it can be represented as a convex combination of the extreme points plus a non-negative linear combination of extreme directions (if there is any). Theorem 12 extends this result to infinite dimensional vector spaces, allowing us to establish the existence of extreme points, and therefore a non-empty feasible region for certain infinite-dimensional linear programming problems. Note that in order to prove attainability of Equalized Odds, it is sufficient to show that the corresponding infinite-dimensional linear programming problem yields a non-empty feasible region. The establishment of duality is relatively complicated for infinite-dimensional linear programming (Anderson and Nash, 1987) and is beyond the scope of the discussion.

**Theorem** *(Attainability of Equalized Odds for Regression with Stochastic Prediction)*
*Let $A$, $X$, and $Y$ be continuous variables with domain of value $\mathcal{A}$, $\mathcal{X}$, and $\mathcal{Y}$, respectively. Assume that their joint distribution is fixed and known. Further assume $Y \not\perp\!\!\!\perp A$, $Y \not\perp\!\!\!\perp X$, and $Y \not\perp\!\!\!\perp X \mid A$. Without loss of generality let the conditional probability density $p_{X|AY}(x \mid a, y)$ be non-negative and finite. Then there exists $\widetilde{Y}$ with domain of value $\mathcal{Y}$ whose distribution is fully determined by $p_{\widetilde{Y}|AX}(\tilde{y} \mid a, x)$, such that $\widetilde{Y}$ is not independent from $(A, X)$ but $\widetilde{Y} \perp\!\!\!\perp A \mid Y$, i.e., the Equalized Odds is non-trivially attainable.*

**Proof** Let us denote $h : \mathcal{A} \times \mathcal{X} \times \mathcal{Y} \to \mathbb{R}$ as the fixed function satisfying $h(a, x, y) = p_{X|AY}(x \mid a, y)$. We want to show that there exists a function $f : \mathcal{A} \times \mathcal{X} \times \mathcal{Y} \to \mathbb{R}$ characterizing the conditional probability density $f(a, x, \tilde{y}) = p_{\widetilde{Y}|AX}(\tilde{y} \mid a, x)$ that satisfies the following conditions:

  (i)  $\forall a \in \mathcal{A}, x \in \mathcal{X}, \tilde{y} \in \mathcal{Y}, f(a, x, \tilde{y})$ is non-negative and finite;
  (ii)  $\forall a \in \mathcal{A}, x \in \mathcal{X}, \int_{\mathcal{Y}} f(a, x, \tilde{y})dt = 1$;
  (iii)  let $Q(a, y, \tilde{y}) := \int_{\mathcal{X}} f(a, x, \tilde{y})h(a, x, y)dx$,
       and then $\forall y, \tilde{y} \in \mathcal{Y}, a, a' \in \mathcal{A}, a \neq a', Q(a, y, \tilde{y}) = Q(a', y, \tilde{y})$;
  (iv)  the function $f(a, x, \tilde{y})$ cannot be written into a function of only $\tilde{y}$.

Conditions (i) and (ii) guarantee that $f$ corresponds to a valid conditional probability density for some $\widetilde{Y}$ conditioned on $A$ and $X$. Condition (iii) is the necessary and sufficient condition for Equalized Odds to hold true for regression tasks (as we have seen in Theorem 2). Condition (iv) guarantees that the corresponding $\widetilde{Y}$ is not independent from $(A, X)$ – otherwise $\widetilde{Y}$ will be a trivial prediction.

Through the lens of functional analysis, we view $f$ as a point in the infinite-dimensional vector space. In order to prove the existence of such a function $f$, we begin by showing that condition (i) and (iii) form a set of constraints of an infinite-dimensional linear programming problem, and that the corresponding feasible region is non-empty. Furthermore, we can always find a point (which is a function) in the aforementioned feasible region that satisfies condition (iv). Finally, condition (ii) is satisfied by "normalizing" over $\tilde{y} \in \mathcal{Y}$.

To begin with, any function that satisfies condition (i) is a bounded function defined on the set $\mathcal{A} \times \mathcal{X} \times \mathcal{Y}$. $\mathbb{R}$ with absolute value norm forms a Banach space. By Lemma 10, the functions that satisfy condition (i) form a Banach space with the supremum norm $\mathcal{F} = (\mathcal{F}, \|\cdot\|_\infty)$. Recall that we assume $h : \mathcal{A} \times \mathcal{X} \times \mathcal{Y} \to \mathbb{R}$ satisfying $h(a, x, y) = p_{X|AY}(x \mid a, y)$ (which is fixed and known) is non-negative and finite. Condition (iii) specifies a set of equality constraints, each of which is characterized with a linear combination of points $f$ in the space $\mathcal{F}$. Therefore conditions (i) and (iii) form a set of constraints of an infinite-dimensional linear programming problem (since we focus on the feasible region of such problem, the exact form of the objective function is omitted). Since for any function $f \in \mathcal{F}$ that can be written into a function of the form $g : \mathcal{Y} \to \mathbb{R}$, $f$ trivially satisfies condition (iii), one can easily construct a non-empty compact subset $\mathcal{K} \subset \mathcal{F}$ such that each point $f \in \mathcal{K}$ satisfies conditions (i) and (iii). In other words, the infinite-dimensional linear programming problem has a non-empty convex feasible region, which contains $\mathcal{K}$.

We now prove that we can find a point in this feasible region that satisfies condition (iv) by showing that any point that violates condition (iv) cannot be an extreme point (as defined in Definition 11) of the feasible region. We have shown that there is a non-empty compact subset $\mathcal{K} \subset \mathcal{F}$, and that $\mathcal{F}$ is a Banach space, and therefore also a locally convex Hausdorff topological vector space. This also implies that $\mathcal{K}$ is closed (since it is a compact subset of a Hausdorff space). By the Krein-Milman Theorem (Theorem 12), the set of extreme points for $\mathcal{K}$ is non-empty. Notice that for any $f_0 \in \mathcal{K}$ that violates condition (iv), $f_0$ can be written into a convex combination of $f_1$ and $f_2$ ($f_1, f_2 \in \mathcal{K}$,

$f_1 \neq f_2$) that also violate condition (iv). For example let $f_i(a, x, \tilde{y}) = g_i(\tilde{y}), i = 0, 1, 2$ (since each $f_i$ violates condition (iv)). Let $g_1(\tilde{y}) = 2 \cdot \mathbb{1}_{(-\infty, 0]}(\tilde{y}) \cdot g_0(\tilde{y})$ and $g_2(\tilde{y}) = 2 \cdot \mathbb{1}_{(0, +\infty)}(\tilde{y}) \cdot g_0(\tilde{y})$ where $\mathbb{1}.(\cdot)$ is the indicator function (without loss of generality assume that the corresponding $f_1, f_2 \in \mathcal{K}$). It is easy to verify that $f_1 \neq f_2$, and $\frac{1}{2} f_1 + \frac{1}{2} f_2 = f_0$. Therefore any $f_0$ that violates condition (iv) cannot be an extreme point of $\mathcal{K}$, which is closed with a non-empty set of extreme points. In other words, there exist a point $f^* \in \mathcal{F}$ such that $f^*$ is an extreme point of $\mathcal{K}$ (therefore $f^*$ is within the feasible region) and that $f^*$ satisfies condition (iv).

Finally, for such $f^*$ within the feasible region, one can find a unique function $\tilde{f} : \mathcal{A} \times \mathcal{X} \times \mathcal{Y}$ such that condition (ii) is satisfied by normalizing over $\tilde{y} \in \mathcal{Y}$ for each possible combination of $(a, x) \in \mathcal{A} \times \mathcal{X}$:

$$\forall a \in \mathcal{A}, x \in \mathcal{X}, \tilde{f}(a, x, \tilde{y}) := \frac{f^*(a, x, \tilde{y})}{\int_{\mathcal{Y}} f^*(a, x, \xi) d\xi}.$$

Therefore, there exists a function $\tilde{f}$ that satisfies conditions (i) to (iv), i.e., Equalized Odds can be non-trivially attained. ∎

### A.4. Proof for Theorem 4

**Theorem** (*Condition to Achieve Equalized Odds for Classification with Deterministic Prediction*) *Assume that the protected feature $A$ and $Y$ are dependent and that their joint probability $P(A, Y)$ (for discrete $A$) or joint probability density $p(A, Y)$ (for continuous $A$) is positive for every combination of possible values of $A$ and $Y$. Further assume that $Y$ is not fully determined by $A$, and that there are additional features $X$ that are not independent of $Y$. Let the output of the classifier $\widehat{Y}$ be a deterministic function $f : \mathcal{A} \times \mathcal{X} \to \mathcal{Y}$. Let $S_A^{(\hat{y})} := \{a \mid \exists x \in \mathcal{X} \text{ s.t. } f(a, x) = \hat{y}\}$, and $S_{X|a}^{(\hat{y})} := \{x \mid f(a, x) = \hat{y}\}$. Equalized Odds is attained if and only if the following conditions hold true (for continuous $X$, replace summation with integration accordingly):*

*(i) $\forall \hat{y} \in \mathcal{Y} : S_A^{(\hat{y})} = \mathcal{A}$,*

*(ii) $\forall \hat{y} \in \mathcal{Y}, \forall a, a' \in \mathcal{A} : \sum_{x \in S_{X|a}^{(\hat{y})}} P_{X|AY}(x \mid a, y) = \sum_{x \in S_{X|a'}^{(\hat{y})}} P_{X|AY}(x \mid a', y)$*

**Proof** We begin by considering the case when $A$ and $X$ are discrete (for the purpose of readability). The Equalized Odds criterion can be written in terms of the conditional probabilities:

$$\begin{aligned} &\forall a \in \mathcal{A}, y, \hat{y} \in \mathcal{Y} : \\ &P_{\widehat{Y}|AY}(\hat{y} \mid a, y) = P_{\widehat{Y}|Y}(\hat{y} \mid y). \end{aligned} \tag{21}$$

Expand the LHS of Equation 21:

$$\begin{aligned} &P_{\widehat{Y}|AY}(\hat{y} \mid a, y) \\ &= \sum_{x \in \mathcal{X}} P_{\widehat{Y}|AXY}(\hat{y} \mid a, x, y) P_{X|AY}(x \mid a, y), \end{aligned}$$

and bear in mind that $\widehat{Y} := f(A, X)$ is a deterministic function of $(A, X)$, we have:

$$\begin{aligned} &P_{\widehat{Y}|AXY}(\hat{y} \mid a, x, y) \\ &= P\big(f(A, X) = \hat{y} \mid A = a, X = x, Y = y\big) \\ &= P\big(f(A, X) = \hat{y} \mid A = a, X = x\big) \in \{0, 1\}. \end{aligned} \tag{22}$$

From Equation 22 we can see that the conditional probability $P_{X|AY}(x \mid a, y)$ can contribute to the summation only when $f(a, x) = \hat{y}$. We can rewrite the LHS of Equation 21:

$$P_{\widehat{Y}|AY}(\hat{y} \mid a, y) = \sum_{x \in S_{X|a}^{(\hat{y})}} P_{X|AY}(x \mid a, y) := Q^{(\hat{y})}(a, y).$$

Similarly, for the RHS of Equation 21, we have:

$$P_{\widehat{Y}|Y}(\hat{y} \mid y)$$
$$= \sum_{a \in \mathcal{A}} \sum_{x \in \mathcal{X}} P_{\widehat{Y}|AXY}(\hat{y} \mid a, x, y) P_{A,X|Y}(a, x \mid y)$$
$$= \sum_{a \in S_A^{(\hat{y})}} \sum_{x \in S_{X|a}^{(\hat{y})}} P_{X|AY}(x \mid a, y) P_{A|Y}(a \mid y)$$
$$= \sum_{a \in S_A^{(\hat{y})}} Q^{(\hat{y})}(a, y) P_{A|Y}(a \mid y).$$

Since Equalized Odds holds true if and only if Equation 21 holds true, then the LHS of the equation does not involve $a$ (as is the case for the RHS), i.e., $Q^{(\hat{y})}(a, y)$ does not change with $a$. Then Equation 21 becomes:

$$Q^{(\hat{y})}(a, y) = \sum_{a \in S_A^{(\hat{y})}} Q^{(\hat{y})}(a, y) P_{A|Y}(a \mid y)$$
$$= Q^{(\hat{y})}(a, y) \sum_{a \in S_A^{(\hat{y})}} P_{A|Y}(a \mid y),$$

which gives condition (i) that $S_A^{(\hat{y})}$ contains all possible values of $A$, i.e., $\mathcal{A} = S_A^{(\hat{y})}$ (otherwise $\sum_{a \in S_A^{(\hat{y})}} P_{A|Y}(a \mid y) < 1$). Since $Q^{(\hat{y})}(a, y)$ does not change with $a$, we have:

$$\forall a, a' \in \mathcal{A}, a \neq a' :$$
$$\sum_{x \in S_{X|a}^{(\hat{y})}} P_{X|AY}(x \mid a, y) = \sum_{x \in S_{X|a'}^{(\hat{y})}} P_{X|AY}(x \mid a', y),$$

which gives condition (ii). Therefore, Equalized Odds implies conditions (i) and (ii). On the other hand, it is easy to see that when conditions (i) and (ii) are satisfied, Equation 21 holds true, i.e., Equalized Odds holds true.

When $A$ and $X$ are continuous, one can replace the summation with integration accordingly. ∎

## A.5. Proof for Theorem 6

**Theorem** (*Attainability of Equalized Odds for Classification with Stochastic Prediction*)
*Assume that the feature $X$ is not independent from $Y$, and that $\widehat{Y}$ is a function of $A$ and $X$. Then for binary classification, if $\widehat{Y}$ is a non-trivial predictor for $Y$, there is always at least one non-trivial (possibly randomized) predictor $\widetilde{Y}_{post}$ derived by post-processing $\widehat{Y}$ that can attain Equalized Odds:*

$$\Omega(\widetilde{Y}_{post}) \neq \emptyset.$$

**Proof** Since $\widehat{Y}$ is a function of $(A, X)$ and $X \not\perp\!\!\!\perp Y$, $\widehat{Y}$ is not conditionally independent from $Y$ given protected feature $A$. Furthermore, since $\widehat{Y}$ is a non-trivial estimator of the binary target $Y$, there exists a positive constant $\epsilon > 0$, such that ($\forall a \in \mathcal{A}$):

$$\left| P_{\widehat{Y}|AY}(1 \mid a, 1) - P_{\widehat{Y}|AY}(1 \mid a, 0) \right| \geq \epsilon. \tag{23}$$

Equation 23 implies that for each value of $A$, the corresponding true positive rate of the non-trivial predictor is always strictly larger than its false positive rate[7]. As illustrated in Panels (a) and (b) of Figure 1, the (FPR, TPR) pair of the predictor $\widehat{Y}$ when $A = a$, i.e., the point $\gamma_a(\widehat{Y})$ on ROC plane, will never fall in the gray shaded area, and its coordinates are bounded away from the diagonal by at least $\epsilon$. Therefore, the intersection of all $\mathcal{C}_a(\widehat{Y})$ would always form a parallelogram with non-empty area, which corresponds to attainable non-trivial post-processing fair predictors $\widetilde{Y}_{\text{post}}$. ∎

### A.6. Proof for Theorem 7

**Theorem** *(Equivalence between ROC feasible areas)*
*et $\Omega(\widetilde{Y}_{post})$ denote the ROC feasible area specified by the constraints enforced on $\widetilde{Y}_{post}$. Then $\Omega(\widetilde{Y}_{post})$ is identical to the ROC feasible area $\Omega(\widetilde{Y}_{in}^*)$ that is specified by the following set of constraints:*
  *(i) constraints enforced on $\widetilde{Y}_{in}$ ;*
  *(ii) additional "pseudo" constraints: $\forall a \in \mathcal{A}$, $\beta_{a0}^{(0)} = \beta_{a1}^{(0)}$, $\beta_{a0}^{(1)} = \beta_{a1}^{(1)}$, where*
  $\beta_{ay}^{(\hat{y})} = \sum_{x \in \mathcal{X}} P(\widetilde{Y}_{in} = 1 \mid A = a, X = x)P(X = x \mid A = a, Y = y, \widehat{Y}_{opt} = \hat{y}).$

**Proof** Since the post-processing predictor $\widetilde{Y}_{\text{post}}$ is derived by optimizing over parameters or functions (of $A$) $\beta_a^{(u)}$. Therefore, considering the fact that $P_{\widetilde{Y}_{\text{post}}|AY}(1|a, y) = \gamma_{ay}(\widetilde{Y}_{\text{post}})$, $P_{\widehat{Y}_{\text{opt}}|AY}(1|a, y) = \gamma_{ay}(\widehat{Y}_{\text{opt}})$, we have the relation between $\gamma_{ay}(\widetilde{Y}_{\text{post}})$ and $\gamma_{ay}(\widehat{Y}_{\text{opt}})$:

$$\gamma_{ay}(\widetilde{Y}_{\text{post}}) = \beta_a^{(0)} \, \gamma_{ay}(1 - \widehat{Y}_{\text{opt}}) + \beta_a^{(1)} \, \gamma_{ay}(\widehat{Y}_{\text{opt}}),$$
$$\beta_a^{(0)} = P(\widetilde{Y}_{\text{post}} = 1 \mid A = a, \widehat{Y}_{\text{opt}} = 0),$$
$$\beta_a^{(1)} = P(\widetilde{Y}_{\text{post}} = 1 \mid A = a, \widehat{Y}_{\text{opt}} = 1).$$

Similarly, consider the relation between positive rates of $\widetilde{Y}_{\text{in}}$ and those of $\widehat{Y}_{\text{opt}}$, i.e., $P_{\widetilde{Y}_{\text{in}}|AY}(1|a, y)$ and $P_{\widehat{Y}_{\text{opt}}|AY}(1|a, y)$, by factorizing $P_{\widetilde{Y}_{\text{in}}|AY}(1|a, y)$ over $X$ and $\widehat{Y}_{\text{opt}}$:

$$P_{\widetilde{Y}_{\text{in}}|AY}(1|a, y) = \sum_{\hat{y} \in \mathcal{Y}} P_{\widehat{Y}_{\text{opt}}|AY}(\hat{y}|a, y) \left[ \sum_{x \in \mathcal{X}} P_{\widetilde{Y}_{\text{in}}|AX}(1|a, x) \cdot P_{X|AY\widehat{Y}_{\text{opt}}}(x|a, y, \hat{y}) \right]. \tag{24}$$

Therefore, we have the relation between $\gamma_{ay}(\widetilde{Y}_{\text{in}})$ and $\gamma_{ay}(\widehat{Y}_{\text{opt}})$:

$$\gamma_{ay}(\widetilde{Y}_{\text{in}}) = \beta_{ay}^{(0)} \, \gamma_{ay}(1 - \widehat{Y}_{\text{opt}}) + \beta_{ay}^{(1)} \, \gamma_{ay}(\widehat{Y}_{\text{opt}}),$$
$$\beta_{ay}^{(0)} = \sum_{x \in \mathcal{X}} P_{\widetilde{Y}_{\text{in}}|AX}(1 \mid a, x)P_{X|AY\widehat{Y}_{\text{opt}}}(x \mid a, y, 0),$$
$$\beta_{ay}^{(1)} = \sum_{x \in \mathcal{X}} P_{\widetilde{Y}_{\text{in}}|AX}(1 \mid a, x)P_{X|AY\widehat{Y}_{\text{opt}}}(x \mid a, y, 1). \tag{25}$$

---

7. If the TPR of the predictor is always smaller than its FPR, one can simply flip the prediction (since the target is binary) and then Equation 23 holds true.

If there is more than one variable in $X$ in Equation 25, one can expand the summation if needed; if some variables are continuous, one may also substitute the summation with integration accordingly.

From Equation 25, $\beta_{ay}^{(0)}$ and $\beta_{ay}^{(1)}$ depend on the value of $Y$:

$$\begin{aligned}
\beta_{ay}^{(0)} &= P(\widetilde{Y}_{\text{in}} = 1 \mid A = a, Y = y, \widehat{Y}_{\text{opt}} = 1), \\
\beta_{ay}^{(0)} &= P(\widetilde{Y}_{\text{in}} = 1 \mid A = a, Y = y, \widehat{Y}_{\text{opt}} = 0).
\end{aligned} \tag{26}$$

Apart from Equalized Odds constraints (which are shared by $\widetilde{Y}_{\text{in}}$ and $\widetilde{Y}_{\text{post}}$), when enforcing additional "pseudo" constraints $\beta_{a0}^{(0)} = \beta_{a1}^{(0)}$ and $\beta_{a0}^{(1)} = \beta_{a1}^{(1)}$, conditional independence $\widetilde{Y}_{\text{in}} \perp\!\!\!\perp Y \mid A, \widehat{Y}_{\text{opt}}$ is enforced, making $\beta_{ay}^{(0)}$ and $\beta_{ay}^{(0)}$ no longer depend on $Y$. This is exactly the inherent constraint $\widetilde{Y}_{\text{post}}$ satisfies. Therefore the stated equivalence between ROC feasible areas $\Omega(\widetilde{Y}_{\text{post}})$ (specified by the constraints enforced on $\widetilde{Y}_{\text{post}}$) and $\Omega(\widetilde{Y}_{\text{in}}^*)$ (specified by the constraints enforced on $\widetilde{Y}_{\text{in}}$ together with the additional "pseudo" constraints) hold true. ■

## Appendix B. Further Information about Experiments

In this section, we first introduce the real-world data sets that are used in the experiments (including regression and classification tasks). Then, we present the technical specifications of the experiments.

### B.1. Description of the Data Sets

(1) **Communities and Crime**: The UCI Communities and Crime data set contains 122 features for 1994 records.[8] The regression task is to predict the number of violent crimes per population for US cities given various census information of the corresponding communities. The data is preprocessed following Mary et al. (2019), and the (continuous) protected feature is the ratio of African-American people in the population.

(2) **Adult** (Kohavi, 1996): The UCI Adult data set contains 14 features for 45,222 individuals (32,561 samples for training and 12,661 samples for testing).[9] The census information includes gender, marital status, education, capital gain, etc. The classification task is to predict whether a person's annual income exceeds 50,000 USD. We use the provided testing set for evaluations and present the result with gender and race (consider white and black people only) set as the protected feature respectively.

(3) **Bank** (Moro et al., 2014): The UCI Bank Marketing data set is related with marketing campaigns of a banking institution, containing 16 features of 45,211 individuals.[10] The assigned classification task is to predict if a client will subscribe (yes/no) to a term deposit. The original data set is very unbalanced with only 4,667 positives out of 45,211 samples. Therefore, we combine "yes" points with randomly subsampled "no" points and perform experiments on the down-sampled data set with 10,000 data points. The protected feature is the marital status of the client.

(4) **COMPAS** (Angwin et al., 2016): The COMPAS data set contains records of over 11,000 defendants from Broward County, Florida, whose risk (of recidivism) was assessed using the

---

8. https://archive.ics.uci.edu/ml/datasets/communities+and+crime
9. http://archive.ics.uci.edu/ml/datasets/Adult
10. https://archive.ics.uci.edu/ml/datasets/bank+marketing

COMPAS tool. Each record contains multiple features of the defendant, including demographic information, prior convictions, degree of charge, and the ground truth for recidivism within two years. Following Zafar et al. (2017a) and Nabi and Shpitser (2018), we limit our attention to the subset consisting of African-Americans and Caucasians defendants. The features we use include age, gender, race, number of priors, and degree of charges. The task is to predict the recidivism of the defendant and we choose race as the protected feature.

(5) **German Credit**: The UCI German Credit data contains 20 features (7 numerical, 13 categorical) describing the social and economical status of 1,000 customers.[11] The prediction task is to classify people as good or bad credit risks. We use the provided numerical version of the data and choose gender as the protected feature.

## B.2. Experimental Setup

### B.2.1. REGRESSION TASKS

The regression experiments consist of two base models, namely, a linear regression model and a neural network model, each of which appears in the form of both deterministic and stochastic prediction. For the neural network model, following Mary et al. (2019) we use a simple regressor for the experiments: two hidden layers (with the same number of neurons ranging from 30 to 100 for each layer, depending on the size of the data set) with SELUs nonlinearity (Klambauer et al., 2017). The network is optimized by minimizing the MSE loss combined with the $\lambda$-weighted fairness penalty term, using the Adam optimizer (Kingma and Ba, 2015). We use the kernel measure of conditional dependence (KMCD) (Fukumizu et al., 2007) as the fairness penalty term (for stochastic prediction), and the weight $\lambda$ ranges from $\{2^{-2}, 2^{-1}, \ldots, 2^{14}\}$. The range of the absolute value of $\lambda$ may differ for different experiment setups since the relative value difference between the MSE loss and the KMCD measure. The learning rate is chosen from $\{10^{-4}, 10^{-5}, 10^{-6}\}$ (smaller learning rate is preferable especially for larger values of $\lambda$, i.e., more emphasis on the fairness penalty), and the batch size is chosen from $\{64, 128, 256\}$. As we have stated in Section 3.3, for stochastic prediction, a random sample from the standard Gaussian distribution is concatenated to each batch of data, and the input dimension of the model changes accordingly. The random sample to concatenate is redrawn each time the data point is seen by the model.

In our experiments on the simulated data, for linear cases, the data is generated as stated in Equation 11, with non-Gaussian distributed exogenous terms ($E_X$, $E_H$, and $E_Y$). Figure 5(a) and (b) correspond to different distributions for the noise terms, specifically, the Laplace distribution and uniform distribution, respectively. We use linear regression with the *Equalized Correlations* constraint (Woodworth et al., 2017), a weaker notion of Equalized Odds for linearly correlated data, as the predictor. For nonlinear cases, the data is generated using a similar scheme but with nonlinear transformations involved and Gaussian distributed exogenous terms. We use a neural network regressor with an Equalized Odds regularization term (Mary et al., 2019) to perform nonlinear fair regression. As we can see in Figure 5(c) and (d), for nonlinear regression tasks, Equalized Odds may not be attained even if every exogenous term is Gaussian distributed.

In our experiments on the real-world data, we perform 20 random splits of *Communities and Crime* data into training ($80\%$) and testing ($20\%$) sets, and present the testing performance for deterministic and stochastic prediction models (for the linear regressor as well as the neural network regressor) in Figure 3. In situations where the protected feature is not available when testing, it is

---

11. https://archive.ics.uci.edu/ml/datasets/statlog+(german+credit+data)

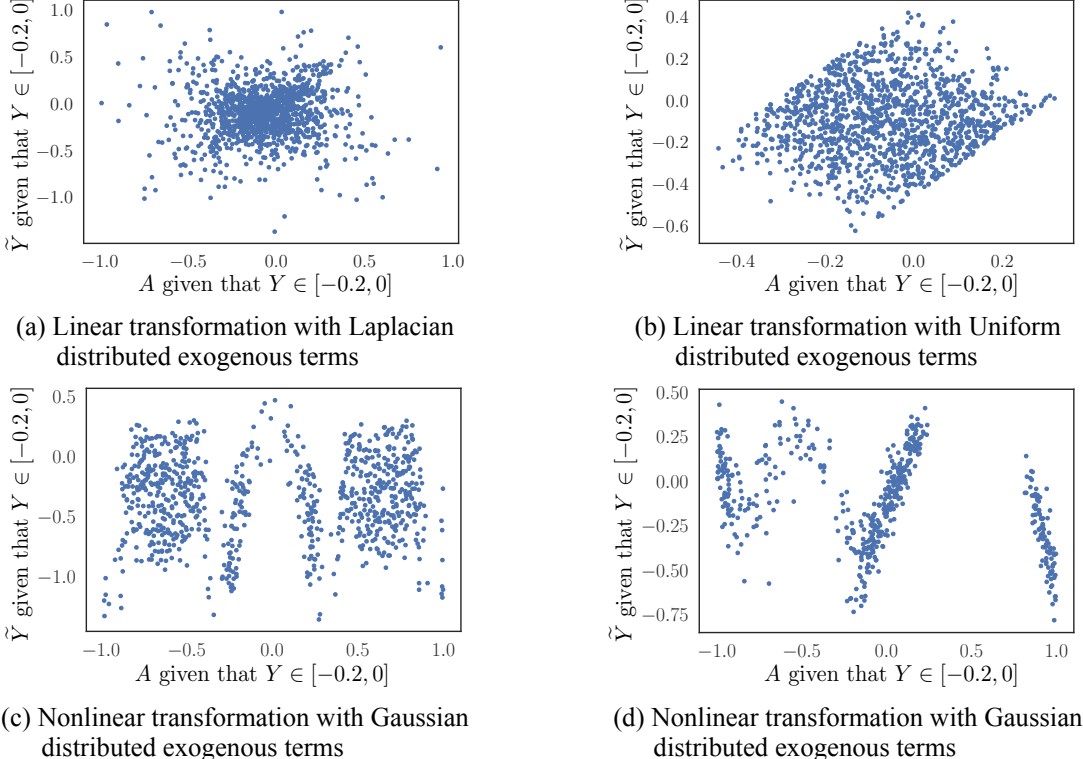

Figure 5: Illustration of unattainable Equalized Odds for regression with deterministic prediction. Panel (a)-(b): Linear regression on the data generated with linear transformations and non-Gaussian distributed exogenous terms (following Laplace, Uniform distribution respectively). Panel (c)-(d): Nonlinear regression with a neural net regressor (Mary et al., 2019) on the data generated with nonlinear transformations and Gaussian exogenous terms. We can observe obvious dependencies between $\widetilde{Y}$ and $A$ on a small interval of $Y$. This indicates the conditional dependency between $\widetilde{Y}$ and $A$ given $Y$, i.e., the Equalized Odds is not achieved.

desirable to exclude the protected feature from the attributes when performing prediction. To this end, during the training process, the input to the regression model does not include the protected feature, and the protected feature is only used to compute the fairness penalty. The related result on the *Communities and Crime* data set is summarized in Figure 6.

### B.2.2. CLASSIFICATION TASKS

In Figure 4, we compare the performance under Equalized Odds of multiple methods proposed in the literature. Hardt et al. (2016) propose a post-processing approach where the prediction is randomized to minimize violation of fairness; Zafar et al. (2017a) use a covariance proxy measure as the regularization term when optimizing classification accuracy; Agarwal et al. (2018) take the reductions approach and reduce fair classification into solving a sequence of cost-sensitive classification problems; Rezaei et al. (2020) minimize the worst-case log loss using an approximated

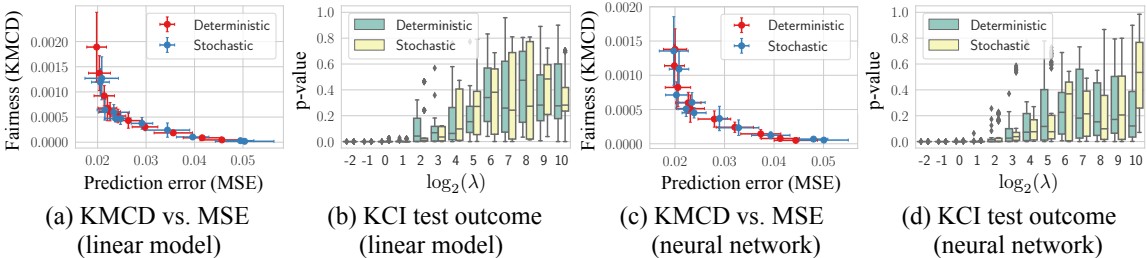

(a) KMCD vs. MSE
(linear model)

(b) KCI test outcome
(linear model)

(c) KMCD vs. MSE
(neural network)

(d) KCI test outcome
(neural network)

Figure 6: Results for regression with deterministic and stochastic prediction functions on the *Communities and Crime* data set (the protected feature is *not* included as an attribute). The base model could be a linear regressor or a neural network regressor, and the stochastic prediction is achieved by introducing an independent noise input sampled from a standard Gaussian distribution.

regularization term; Baharlouei et al. (2020) propose to use Rényi correlation as the regularization term to account for nonlinear dependence between variables. To measure the violation of the fairness criterion, we use Equalized Odds (EOdds) violation, defined as $\max_{y \in \mathcal{Y}\ a, a' \in \mathcal{A}} \left| P_{\widetilde{Y}|AY}(1|a, y) - P_{\widetilde{Y}|AY}(1|a', y) \right|$. Following Agarwal et al. (2018), we pick 0.01 as the default violation bound that the EOdds violation does not exceed (if practically achievable for the method) during training. For each method we plot the testing accuracy versus the violation of Equalized Odds.

## Appendix C. Additional Derivation and Illustrative Examples

### C.1. Derivation of the Connection between Theorem 2 and Theorem 1

Theorem 2 presents the condition under which Equalized Odds can possibly hold true for regression with deterministic prediction. This condition specifies a rather strong constraint on the relation between the conditional probability density $p_{X|AY}(x \mid a, y)$ and the deterministic function $f : \mathcal{A} \times \mathcal{X} \to \mathcal{Y}$. In order to further illustrate this constraint, let us consider a special case where variables are linear correlated with Gaussian exogenous terms (i.e., $E_X$ and $E = E_Y + dE_H$ in Equation 11 are Gaussian) and $\widehat{Y} = f(A, X)$ is a linear function of $(A, X)$ with linear coefficients $\alpha$ and non-zero $\beta$ (i.e., $\widehat{Y} = \alpha A + \beta X$). Here, the conditional distribution of $X$ given $A$ and that of $Y$ given $A$ are both Gaussian. Then for any given value of $(\widehat{Y}, A)$ we have $X = (\widehat{Y} - \alpha A)/\beta$. Therefore $\delta(\hat{y} - f(a, x))$ does not equal to 0 only on the singleton set $\{x \mid f(a, x) = \hat{y}\} = \{(\hat{y} - \alpha a)/\beta\}$. We have

$$Q(a, y, \hat{y}) = p_{X|AY}(\frac{\hat{y} - \alpha a}{\beta} \mid a, y).$$

Then the condition in Theorem 2 requires that

$$p_{X|AY}(\frac{\hat{y} - \alpha a}{\beta} \mid a, y) = p_{X|AY}(\frac{\hat{y} - \alpha a'}{\beta} \mid a', y). \tag{27}$$

Since the conditional distribution of $X$ given $A = a$ and that of $Y$ given $A = a$ are both Gaussian ($\mathcal{N}(qa, \sigma_X^2)$, and $\mathcal{N}((qc + bd)a, c^2\sigma_X^2 + \sigma_E^2)$ respectively), we have $p_{X|AY}(x \mid a, y) = \frac{1}{\sqrt{2\pi\sigma_{ay}^2}} \exp\{\frac{(x-\mu_{ay})^2}{2\sigma_{ay}^2}\}$, where $\mu_{ay} = qa + \frac{c\sigma_X^2}{c^2\sigma_X^2+\sigma_E^2}(y - (qc + bd)a)$, $\sigma_{ay}^2 = (1 - \frac{c^2\sigma_X^2}{c^2\sigma_X^2+\sigma_E^2})\sigma_X^2$.

Then the Equation 27 hold true if and only if the absolute distance between $\frac{\hat{y}-\alpha a}{\beta}$ and $\mu_{ay}$ is not a function of $a$ (for any fixed $y$ and $\hat{y}$), which yields the following relation:

$$\frac{\alpha}{\beta} = \frac{bdc \cdot \sigma_{E_X}^2 - q \cdot \sigma_E^2}{c^2 \cdot \sigma_{E_X}^2 + \sigma_E^2},$$

which is the exact relation between $\alpha$ and $\beta$ derived from a different perspective in Corollary 9, whose original proof builds up on top of Theorem 1.

As we can see in this example, in order to achieve Equalized Odds, the coefficients of linear function $f(A, X) = \alpha A + \beta X$ have to satisfy a very specific relation. This indicates the general restrictiveness of the aforementioned constraint on the relation between $p_{X|AY}(x \mid a, y)$ and the deterministic function $f$, i.e., Equalized Odds cannot hold true.

## C.2. Deterministic Classification Examples

Theorem 4 specifies the conditions for Equalized Odds to hold true for deterministic classification. Here we present concrete examples where those conditions can or cannot be satisfied. For the purpose of simplifying the notation, let us consider cases when $A$, $X$, and $Y$ are binary, and the joint probability of $(A, Y)$ is specified as following:

$$\begin{aligned}
P(A = 0, Y = 0) &= 0.2, \\
P(A = 0, Y = 1) &= 0.4, \\
P(A = 1, Y = 0) &= 0.3, \\
P(A = 1, Y = 1) &= 0.1.
\end{aligned} \tag{28}$$

Then in the special case when $X \perp\!\!\!\perp A \mid Y$, Equalized Odds can hold true if $f$ is a function of only $X$. For example, if $f = \mathbb{1}\{X = 1\}$, one can quickly verify that $P(\widetilde{Y} = 1 \mid A = a, Y = 0) = P(X = 1 \mid Y = 0)$ and that $P(\widetilde{Y} = 1 \mid A = a, Y = 1) = P(X = 1 \mid Y = 1)$ (for $a \in \{0, 1\}$), i.e., Equalized Odds holds true.

If $X \not\perp\!\!\!\perp A \mid Y$, let us compare the following two cases:

$$\begin{aligned}
P(X = 1 \mid A = 0, Y = 0) &= 0.3, \\
P(X = 1 \mid A = 1, Y = 0) &= 0.7, \\
P(X = 1 \mid A = 0, Y = 1) &= 0.8, \\
P(X = 1 \mid A = 1, Y = 1) &= 0.2;
\end{aligned} \tag{29}$$

$$\begin{aligned}
P(X = 1 \mid A = 0, Y = 0) &= 0.4, \\
P(X = 1 \mid A = 1, Y = 0) &= 0.7, \\
P(X = 1 \mid A = 0, Y = 1) &= 0.6, \\
P(X = 1 \mid A = 1, Y = 1) &= 0.2.
\end{aligned} \tag{30}$$

Here, the joint distribution of $(A, Y)$ is specified the same as in Equation 28. In Equation 29, the conditional probability of $X$ is crafted so that $P(X = 0 \mid A = 0, Y = 0) = P(X = 1 \mid A = 1, Y = 0)$ and $P(X = 0 \mid A = 0, Y = 1) = P(X = 1 \mid A = 1, Y = 1)$. Then if we choose $f = \mathbb{1}\{A = X\}$ (or flip the prediction, let $f = 1 - \mathbb{1}\{A = X\}$) to exploit this property, we can have a predictor that satisfies Equalized Odds. One can quickly verify this: $P(\widetilde{Y} = 1 \mid A = 0, Y = 0) =$

Table 1: Illustration of the benefit of introducing stochastic prediction in a binary classification example. Here we tune the hyperparameter such that the fairness violation is as small as possible for the deterministic predictor. As we can see, given a similar accuracy level for predictors optimized with fairness consideration (Equalized Odds), the stochastic classifier has a much smaller empirical violation of fairness compared to its deterministic counterpart.

|  | Base model | Deterministic classifier | Stochastic classifier |
|---|---|---|---|
| Accuracy | $0.71 \pm 0.01$ | $0.55 \pm 0.02$ | $0.54 \pm 0.03$ |
| (Empirical) Fairness violation | $1.0 \pm 0.00$ | $0.34 \pm 0.04$ | $0.03 \pm 0.01$ |

$P(X = 0 \mid A = 0, Y = 0) = 0.6 = P(X = 1 \mid A = 1, Y = 0) = P(\widetilde{Y} = 1 \mid A = 1, Y = 0)$, and $P(\widetilde{Y} = 1 \mid A = 0, Y = 1) = P(X = 0 \mid A = 0, Y = 1) = 0.2 = P(X = 1 \mid A = 1, Y = 1) = P(\widetilde{Y} = 1 \mid A = 1, Y = 1)$. However, there is no obvious reason to believe that the conditional probability of $X$ given $A$ and $Y$ should satisfy such "crafted" property in general cases. In the case shown in Equation 30, one cannot find an $f$ that satisfies conditions (i) and (ii) in Theorem 4, and therefore in this case there is no deterministic prediction function that satisfies Equalized Odds.

### C.3. Benefit of Introducing Stochastic Prediction under Almost-the-Same Performance

To illustrate the benefit of utilizing stochastic predictors, we compare deterministic and stochastic prediction schemes in terms of their quantitative violations of fairness, given almost the same prediction performance, in several specific classification and regression scenarios.

#### C.3.1. A BINARY CLASSIFICATION EXAMPLE

We generate simulated data according to the joint distribution specified by Equation 28 and Equation 30 with a sample size of 500, and use logistic regression as our base model. We utilize the deterministic predictor proposed by Baharlouei et al. (2020) and the stochastic predictor proposed by Agarwal et al. (2018), and numerically compare the fairness violation (here is the maximum difference between True/False Positive Rates across groups) given almost the same accuracy. We summarize the results of 10 repetitive experiments in Table 1.

#### C.3.2. A LINEAR NON-GAUSSIAN REGRESSION EXAMPLE

We generated the simulated data according to Equation 11, where $q = 0.7$, $b = 0.6$, $c = 0.9$, and $d = 0.6$. We use the same set of nonlinear deterministic and stochastic predictors as we specified in Section B.2.1. Similar to the previous example, we summarize the results for 10 repetitive experiments. For the purpose of demonstrating the influence from characteristics of the data itself, we present simulation results with various exogenous terms that are generated from different non-Gaussian distributions in Table 2 (Uniform distribution) and Table 3 (Laplace distribution). Because the violation of fairness for regression is much easier observed via hypothesis test results, we present the frequency of p-value (for KCI test) exceeding 0.05 threshold (the more often, the more probable the predictor is fair). We can see that given a similar MSE level, the stochastic regressors are more probable to be fair compared to the deterministic ones.

Table 2: Illustration of the benefit of introducing stochastic prediction in regression. Here the exogenous terms in Equation 11 satisfy $E_X \sim U[-0.2, 0.2]$, $E_H \sim U[-0.2, 0.2]$, and $E_Y \sim U[-0.1, 0.1]$.

|  | Base model | Deterministic regressor | Stochastic regressor |
|---|---|---|---|
| MSE | $0.003 \pm 0.001$ | $0.059 \pm 0.004$ | $0.058 \pm 0.007$ |
| How often: p-value $> 0.05$ | 0 out of 10 | 1 out of 10 | 8 out of 10 |

Table 3: Illustration of the benefit of introducing stochastic prediction in regression. Here the exogenous terms in Equation 11 satisfy $E_X \sim \mathrm{Laplace}(0, 0.4)$, $E_H \sim \mathrm{Laplace}(0, 0.4)$, and $E_Y \sim \mathrm{Laplace}(0, 0.2)$.

|  | Base model | Deterministic regressor | Stochastic regressor |
|---|---|---|---|
| MSE | $0.004 \pm 0.001$ | $0.061 \pm 0.003$ | $0.061 \pm 0.007$ |
| How often: p-value $> 0.05$ | 0 out of 10 | 0 out of 10 | 9 out of 10 |

## Appendix D. An Example of Unwanted Usage of Stochastic Prediction

For classification, while randomization can ensure group level of fairness, there is still some inherent shortcoming of the criterion that we should pay attention to. For example, in the FICO case study in Hardt et al. (2016), for a specific client from certain demographic group, the decision of approve/deny the loan actually comes in two folds: if his/her credit score is above (below) the upper (lower) threshold, the bank approve (deny) the application for sure; if the score falls in the interval between two thresholds, the bank would flip a coin to make a decision. Then we can imagine the following situation when a client whose credit score falls within the interval between the upper and lower thresholds goes to a bank to apply a loan. He/she can ask (if conditions permit) the bank to run the model multiple times until the decision is approval. This would make the randomization that was built into the system for the sake of fairness no longer effective. The system in fact only has one fixed threshold (i.e., the original lower threshold), which is in essence a deterministic predictor (therefore in general, does not satisfy Equalized Odds).

