# OpenReview forum: "Attainability and Optimality: The Equalized Odds Fairness Revisited"
_cclear.cc/CLeaR/2022/Conference — CLeaR 2022 Oral_

### Official Review · Reviewer_URgY · 2021-11-26

**Confidence:** 3
**Overall Score:** 7

**Main Review:**

The authors focus on the Equalized Odds notion of fairness and discuss its attainability for deterministic as well as stochastic prediction functions. The message of the paper is that much stronger conditions are required for attainability of Equalized odds when the prediction function is deterministic. The authors also compare the performance of a fair predictor derived via an in-processing approach compared to a post-processing approach, and show that the former generally performs better. For the most part, the paper is well-written and easy to follow. I believe the results are interesting and novel, but it is not clear that how it should be used in practice. The main concern about the results of this work is that the authors discuss whether equalized odds is realized or not, but they do not quantify its violation. For instance, if a class of predictors cannot achieve equalized odds but (based on a reasonable metric) it almost satisfies it, considering the fact that in reality we always work with finite data, this should not be a real concern. The authors suggest that in general using stochastic prediction functions is preferred in terms of obtaining equalized odds notion of fairness but introducing randomness to the predictor may have other negative side effects such as reducing the accuracy or issues mentioned by the authors in Subsection 6.2. Doing the trade-off is only meaningful when both sides are quantified. In other words, it is not exactly clear how much we are gaining in terms of fairness and what we are trading it for if we restrict ourselves to a specific class of stochastic prediction functions.

**Summary:**

---

### Official Review · Reviewer_dWfh · 2021-11-26

**Confidence:** 2
**Overall Score:** 7

**Main Review:**

The fairness of the algorithm is an important topic, especially in machine learning.\
The paper builds on prior work in the field of equalized odds fairness. The authors put some effort into finding the conditions under which equalized odds can hold or not for regression and classification tasks.\
The theoretical analysis or experimental results are presented in a logical way. Unfortunately, I can not find the Appendix in Supplementary Material and read more details for their proofs and experiments. Do I miss something? Besides, the authors also show the limitation of their work.\
This paper is well-written and well-organized. \
Overall, their conclusions are both novel and interesting. I will tend to accept this paper.

I am unfamiliar with this topic, so I will change my score if the other reviewers provide some new points.

**Summary:**

This paper considers the fairness problem of machine learning algorithms, especially focusing on the equalized odds fairness. The main contribution is that the authors discuss the attainability of fairness on two essential tasks, i.e., regression and classification. The authors show that though some settings may not satisfy equalized odds, one may obtain the necessary and sufficient conditions under which the equalized odds can hold.

---

### Official Review · Reviewer_aHTR · 2021-11-26

**Confidence:** 2
**Overall Score:** 6

**Main Review:**

Originality: the originality of the paper consists in showing that for regression and classification (under determinism) Equalized Odds is in general not attainable. It is also shown that with a stochastic predictor, under mild assumptions, one can always derive an Equalized Odds predictor.

Significance: the paper addresses for sure a very important problem, but I do not see much relevance for causal learning.

Technical quality: the proposed approach is sound. Claims are well substantiated.

Clarity: the paper is well written and organised. However, the different notions of algorithmic fairnesses should have been discussed and illustrated with examples.

**Summary:**

The paper is about algorithmic fairness. The main contribution of the paper is to show that Equalized Odds proposed by Hardt et al. (2016) is not always attainable if one uses deterministic prediction functions.

---

### Decision · Program_Chairs · 2022-01-12

**Decision:**

Accept (Poster)

**Comment:**

This paper revisits the attainability of the Equalized Odds definition using a causal view. The paper revealed scenarios when the Equalized Odds definition is in fact unattainable. Conditions for when the Equalized Odds holds true are given too, and for these cases, the paper further studies the optimality of the prediction performance.

The reviewers are unanimously positive about the paper's technical contribution and its relevance to CLeaR.